

# Effects of turbulence structure and urbanization on the heavy haze pollution process

Yan Ren[1], Hongsheng Zhang[1], Wei Wei[2], Bingui Wu[3], Xuhui Cai[4], Yu Song[4]

[1]Laboratory for Climate and Ocean-Atmosphere Studies, Department of Atmospheric and Oceanic Sciences, School of Physics, Peking University, Beijing 100081, P.R. China
[2]State Key Laboratory of Severe Weather, Chinese Academy of Meteorological Sciences, Beijing 100081, P.R. China
[3]Tianjin Municipal Meteorological Bureau, Tianjin 300074, P.R. China
[4]State Key Joint Laboratory of Environmental Simulation and Pollution Control, Department of Environmental Science, Peking University, Beijing 100081, P.R. China

*Correspondence to*: Hongsheng Zhang (hsdq@pku.edu.cn)

**Abstract.** In this paper, an automated algorithm, which is used to identify the spectral gap during the heavy haze pollution process, reconstruct acquired data, and obtain pure turbulence data, is developed. Comparisons of the reconstructed turbulent flux and eddy covariance (EC) flux show that there are overestimations regarding the exchange between the surface and the atmosphere during heavy haze pollution episodes. After reconstruction via the automated algorithm, pure turbulence data can be obtained. We introduce a definition to characterize the local intermittent strength of turbulence (LIST). The trend in the LIST during pollution episodes shows that when pollution is more intense, the LIST is smaller, and intermittency is stronger; when pollution is weaker, the LIST is larger, and intermittency is weaker. During the heavy pollution process, air quality on an hourly scale is determined via turbulent motion. At the same time, the LIST at the city site is greater than that at the suburban site, which means that intermittency over the complex city area is weaker than that over the flat terrain area. Urbanization seems to reduce intermittency during heavy haze pollution episodes, which means that urbanization reduces the degree of weakening in turbulent exchange during pollution episodes. This result is confirmed by comparing the average diurnal variations in turbulent fluxes at urban and suburban sites during polluted and clean periods. The sensible heat flux, latent heat flux, momentum flux, and turbulent kinetic energy (TKE) in urban and suburban areas are all affected when pollution occurs. Material and energy exchanges between the surface and the atmosphere are inhibited. Moreover, the impact of the pollution process on suburban areas is much greater than that on urban areas. Urbanization seems to help reduce the consequences of pollution.

## 1 Introduction

PM$_{2.5}$ (fine particular matter, with a diameter smaller than 2.5 μm) has become the foremost pollution issue in China (Zhang et al. 2012; Hu et al., 2014), particularly in Beijing and its vicinity, in terms of severity and duration (Watts, 2005; Tang et al., 2009, 2012, 2015; Zhang, et al. 2014; Yang et al.,2015;). Because of its negative impacts on human health (Dominici et al. 2014; Thompson et al., 2014) and complicated effects on weather and climate (Liepert et al., 2004; Wang et al., 2010;



Zhang et al., 2015; Wang et al., 2015), heavy aerosol pollution has prompted the close attention of not only scientists but also government and everyday people.

During pollution periods, previous studies have indicated that meteorological conditions always include constant stagnant winds, relatively increased water-vapor density, and strong stable stratification, which inhibits the diffusion of vertical pollution and results in the explosive growth of $PM_{2.5}$ in Beijing (Zhang et al., 2015; Wei et al. 2018; Zhong et al. 2018). The diurnal variations in particulate concentrations are mostly dominated by the diurnal variability of the boundary layer and source emissions (Liu et al., 2014); Turbulent motion determines the meteorological elements in the atmospheric boundary layer, such as turbulence diffusion, PBL height and atmospheric circulation patterns, are all key to hazy weather (Wang et al., 2015; Tang et al., 2016; Zhu et al., 2018; W. Zhang et al., 2016;) and dominate whether the haze occurs or not, since emissions can remain stable within a defined period in a certain area. Therefore, the spatial and temporal characteristics of turbulent activity have significant effects on the local air quality from hourly to diurnal scales under a stable boundary layer (Salmond and Kendry, 2005).

Turbulence in the stable boundary layer is weak and typically characterized by intermittent turbulence or even no turbulence at a variety of heights, temporal scales and spatial locations (Mahrt, 1998; Coulter and Doran, 2002; Van de Wiel et al., 2003; Salmond, 2005; Mahrt, 2014). The term intermittency has different meanings that vary among studies (Coulter and Doran, 2002; Muschinski et al., 2004; Acevedo et al., 2006; Mahrt, 2007). Mahrt (1989, 1999) defines intermittency as the case where eddies at all scales are missing or suppressed at scales that are large compared to those for large eddies. A number of studies have indicated that intermittency is driven by nonstationarity due to motion at time scales that are slightly greater than those for turbulence (Mahrt, 2007; Mahrt, 2010) when the large-scale flow is weak. These motions are sometimes referred to as submesoscale motions (Nappo, 2002; Sun et al., 2004; Anfossi et al., 2005; Conangla et al., 2008; Mahrt et al., 2008). Some studies have simply defined intermittency as "another name for nonstationary" (Treviño, 2000).

As mentioned earlier, the occurrence of heavy haze is always accompanied by low wind speeds and strong stable stratification. Under these conditions, exchanges between the surface and the atmosphere cannot be calculated with the eddy-correlation technique when intermittent turbulence exists because of the nonstationarity imposed by submesoscale motions (Vickers and Mahrt, 2006; Acevedo et al., 2006, 2007; Aubinet, 2008; Mahrt, 2010), which makes it very difficult to accurately determine the transport, storage and diffusion of pollutants, particularly in regions with complex terrain (Bowen et al., 2000), such as Beijing. Densely constructed buildings on the underlying surface aggravate the complexity of turbulent exchange during pollution periods, and the impact of urbanization on pollution is unknown. The Monin-Obukhov similarity theory establishes the relationship between turbulent flux and the vertical gradient in the surface layer and has been widely used in weather and climate models (Wood et al., 2010; Wilson, 2008). However, turbulent flux calculated by the traditional time-averaging method is contaminated by submesoscale motions during the heavy pollution process, which makes the application of similar theories difficult and may be the reason that the simulated pollutant concentration is lower during the heavy pollution process (Li et al., 2016). To provide a basis for improving the parameterization scheme of the atmospheric





pollution diffusion model, it is very important to accurately calculate pure turbulent transport during the heavy pollution process.

Therefore, the purpose of this study is to separate classic turbulent motions from submesoscale motions. At the same time, intermittency can be defined by the turbulent and nonturbulent portions of a signal once the criteria for identifying the

boundary between them have been established (Salmond, 2005). Only in this way can we identify the classic turbulence exchange during pollution periods, regardless of whether turbulence occurs over a flat underlying surface or a complex underlying surface. In a previous study, Muschinski (2004) found a plateau in the intermittency spectra. Vickers and Mahrt (2003) showed the existence of a cospectral gap, which separated turbulent and mesoscale contributions to calculate the fluxes in heat, moisture, and momentum used the multiresolution decomposition technique. Salmond (2005) used the

wavelet analysis to objectively isolate intermittent turbulent bursts within vertical velocity time series. Wei et al. (2017) found that the spectral gap separates fine-scale turbulence from large-scale motions using the arbitrary-order Hilbert spectral analysis during intermittency in the Cooperative Atmosphere–Surface Exchange Study-1999 (CASE-99). Because of the nonstationarity and nonlinearity of intermittent turbulence, analytical techniques have significant impacts on the results. A new technique, named the arbitrary-order Hilbert spectral analysis (Huang et al., 2011), showed its advantage and validity in

the application of turbulent flow and intermittency (Wei et al. 2016, 2017, 2018).

In this paper, we use the arbitrary-order Hilbert spectral method to analyze turbulence data observed from several severe haze pollution episodes that occurred in Beijing and its nearby suburbs from 16 December 2016 to 8 January 2017 to identify the spectral gap and separate classic turbulence from the original signal. After obtaining the pure turbulence signal and the strength of the submesoscale motions, the classic turbulent flux and intermittency strength can be calculated. Then, we can

obtain the macrostatistical characteristics of turbulence over a flat terrain in suburban Beijing and discover the difference between clean and polluted days. This analysis is helpful for improving the current understanding of the transport and diffusion of $PM_{2.5}$ and the intermittency strength of turbulence when heavy haze pollution occurs. The effect of urbanization on the heavy pollution process is simultaneously studied by comparing the results of the urban and suburban sites.

## 2 Description of the sites and data

### 2.1 Experimental site

The data used in this paper are from two measurement sites (Fig. 1 shows the location of the two sites). Data over a flat terrain were collected at a continuous measurement site (40°16′N, 116°28′E) in a Beijing suburb. The observational site was set up in the middle of a vast and horizontal farmland. A 10-m tower was erected for the eddy covariance (EC) and meteorological measurements. The EC system was mounted at a height of 8.3-m above ground. The system was equipped

with an integrated $CO_2/H_2O$ open-path gas analyzer and three-dimensional sonic anemometer (IRGASON, Campbell Scientific, Inc., USA). The IRGASON was leveled and pointed north. The turbulence data were collected using a data logger



(CR3000, Campbell Scientific, Inc., USA) at a frequency of 10 Hz. For convenience of description, this site is described as the suburban site.

The second dataset was collected over urban terrain at a continuous measurement site at Peking University (39°99′N, 116°31′E) in Beijing, which is the capital of China. The instruments are situated on top of a building at the School of Physics

(Peking University) and extend 25-m above ground. There are teaching and residential buildings and public transit facilities that surround the station. The site is located in a typical urban landscape. Details on the site introduction and data background have been shown in Ren et al. (2018). The concentrations of $PM_{2.5}$ were collected using a Thermo-Fisher Sci. Co. instrument (series FH-62-C14), and 30-min averaging time series were performed to remove outliers. Wind vectors and temperature fluctuations were logged at 10 Hz using a 3D sonic anemometer (Campbell Co., USA). For convenience of

description, this site is described as the urban site.

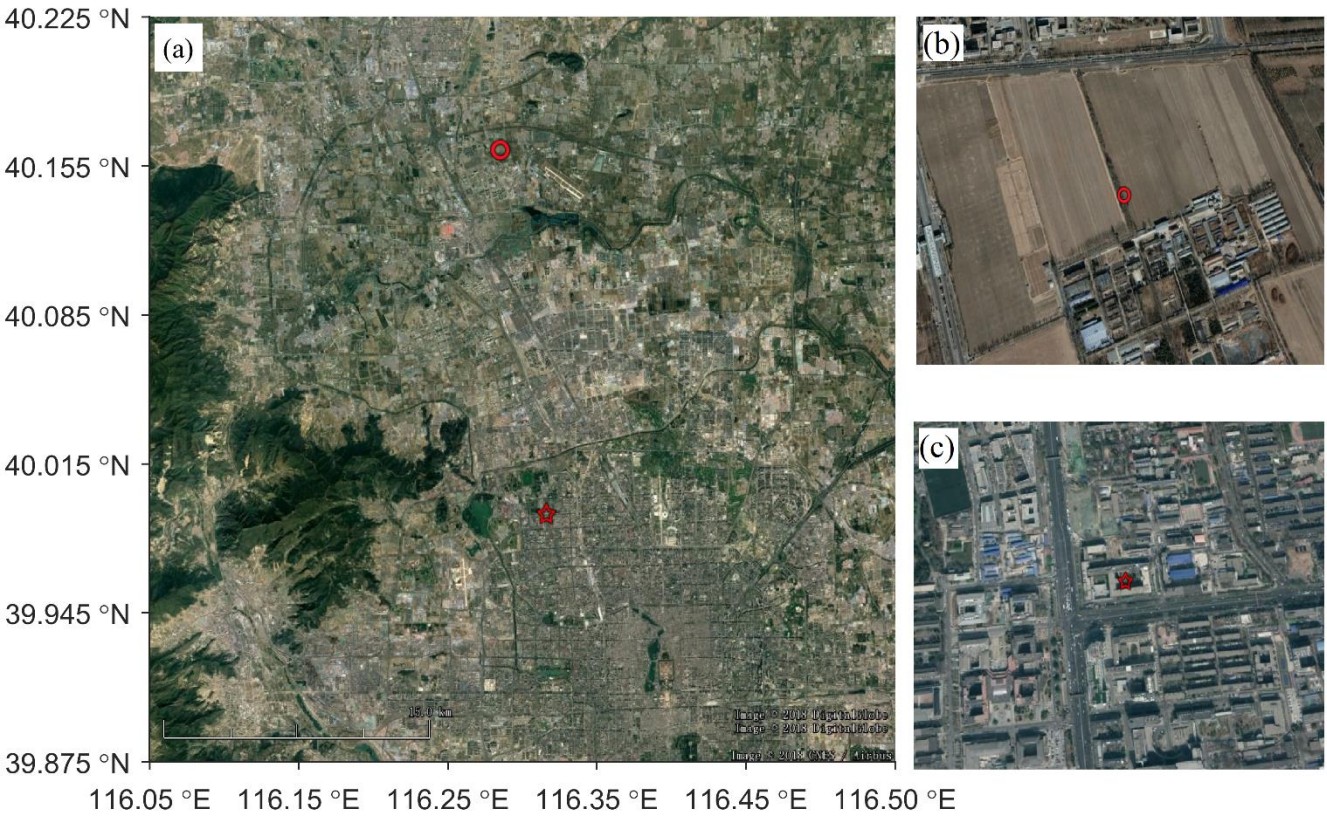

**Figure 1: Google Earth map of the observation sites in Beijing: (a) the observation site located in the urban underlying surface region (marked by the red pentagram) and the observation site located in the suburban underlying surface region with a flat terrain (marked by the red concentric circle). The corresponding terrains (within a range of approximately 500 m) around the**

**observation sites are shown in (b) and (c), respectively.**



## 2.2 Data processing

In this study, we select a 24-day pollution period from 16 December 2016 to 8 January 2017; the number of heavy pollution days with PM$_{2.5}$ concentrations exceeding 200 μg m$^{-3}$ reached 15 (17-21, 24-25, 28, and 30-31 December 2016 and 1-7 January 2017). The remaining days were identified as clear days. The PM$_{2.5}$ mass concentration, horizontal wind speed, virtual temperature and water vapor mixing ratio are shown in Fig. 2 during the days of concern. The trends in meteorological elements over time are consistent at both sites. The horizontal wind speed is larger during clear days and smaller during pollution days; the virtual temperature seems to be greater during pollution days, but this result is not obvious. The water-vapor content is significantly higher during pollution days than that during clear days. The characteristics of these meteorological elements during the pollution process are similar to those in previous studies (Wang et al., 2014; Sun et al., 2014; Tai et.al, 2010). The horizontal wind speed, virtual temperature and water vapor at the suburban site are greater, lower, and higher than those at the city site, respectively. These differences between the two sites are caused by the difference in surface type. High-rise buildings across the city landscape weaken the horizontal wind speed.

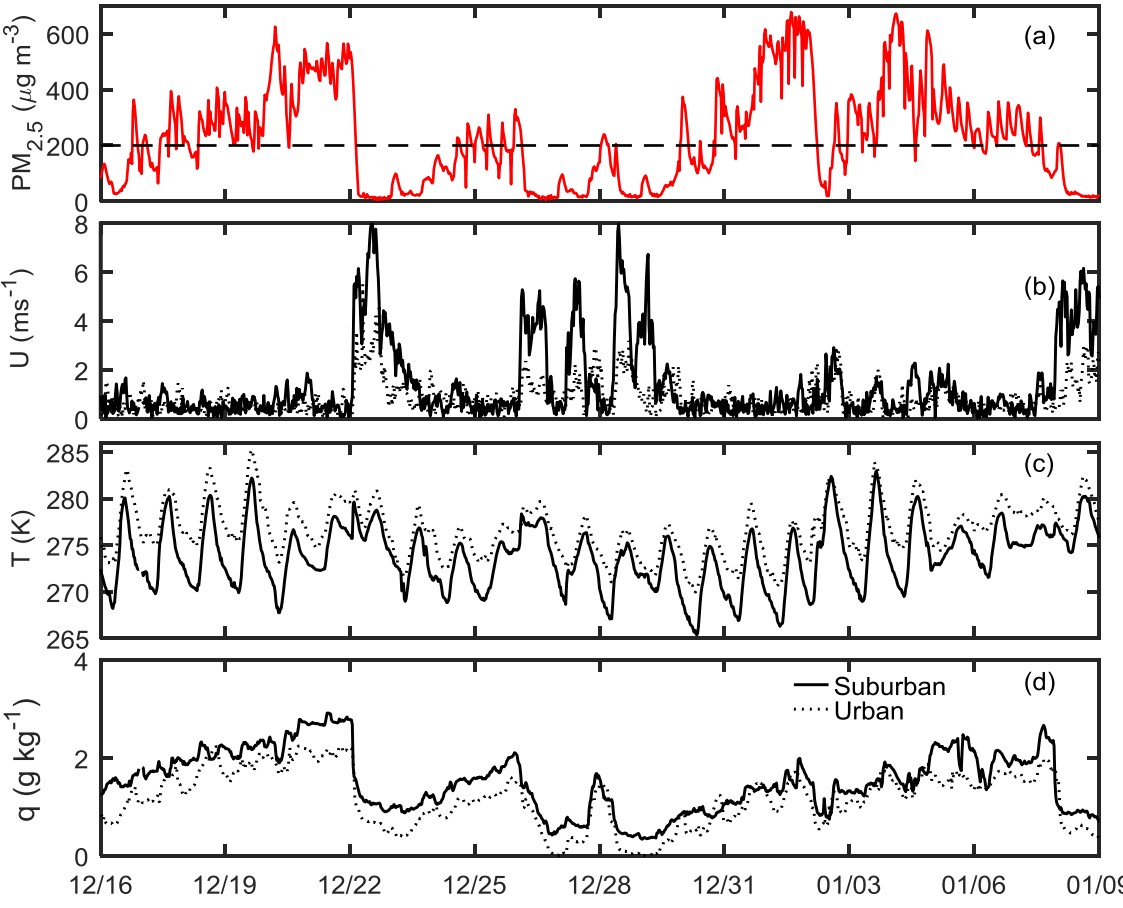



**Figure 2: The PM$_{2.5}$ mass concentration (a), horizontal wind speed (b), virtual temperature (c) and water-vapor mixing ratio (d) from 16 December 2016 to 8 January 2017. The black solid line represents data from the suburbs of Beijing (flat terrain), and the black dotted line represents data from the Peking University site (urban landscape).**

The raw turbulence data were preprocessed over an averaging interval of 30 min by the Eddy Pro software (Advanced 4.2.1, LI-COR Biosciences, Inc., USA). Preprocessing included procedures involving error flags, despiking (following Vickers and Mahrt, 1997), double rotations (following Wiczak et al., 2001), and detrending, which are implemented using the block averaging method. In this paper, the EC method is used to calculate the flux for comparison, where the average time is 30 min. At the same time, strict quality control is performed on the data and data groups to remove data meeting any of the following conditions: (1) the EC method runs with more than $\pm 120\degree$ between the wind direction and direction of the sonic anemometer, (2) the included angle between the wind direction and horizontal plane is greater than $3\degree$, (3) the wind speed is less than 0.5 m s$^{-1}$, (4) the friction speed is less than 0.05 m s$^{-1}$, (5) the sensible heat flux is less than 5 W m$^{-2}$, and (6) data groups that are obviously incorrect. The physical quantities used in this paper are turbulent kinetic energy (TKE), several variances ($\sigma_u$, $\sigma_v$, $\sigma_w$, $\sigma_\theta$ and $\sigma_q$), friction speed ($u_*$), and fluxes ($-\overline{u'w'}$, $\overline{w'\theta'}$ and $\overline{w'q'}$). Among these, the TKE is calculated as:

$$e = \frac{1}{2}\left(\overline{u'^2} + \overline{v'^2} + \overline{w'^2}\right), \tag{1}$$

the variance is calculated as

$$\sigma_u = \overline{u'u'}, \sigma_v = \overline{v'v'}, \sigma_w = \overline{w'w'}, \sigma_\theta = \overline{\theta'\theta'}, \text{ and } \sigma_q = \overline{q'q'}, \tag{2}$$

and the friction speed is calculated as:

$$u_* = \left[\left(-\overline{u'w'}\right)^2 + \left(-\overline{v'w'}\right)^2\right]^{\frac{1}{4}}, \tag{3}$$

## 3 Reconstructed Signals

### 3.1 Automated algorithm to identify the spectral gap

In the traditional turbulence theory, Van der Hoven (1957) presented an analysis on the large spectrum of horizontal wind speeds, which showed a notable spectral gap of approximately 1 cycle h$^{-1}$ between the macroscale (synoptic) and microscale (turbulent) parts of the spectrum. Many other studies have generally proven the existence of the spectral gap (Panofsky, 1969; Fiedler and Panofsky, 1970; Smedman and Hogstrom, 1974). As observations progress, many studies on the stable boundary layer have indicated that the turbulent part of the signal decreases, the time scale for turbulence decreases, and the time scale for the spectral gap also decreases, as mentioned above. These studies used the multiresolution decomposition method to choose a variable averaging time to remove the effects of mesoscale motions. The method we used here to separate pure turbulence and submesoscale motion is based on half-hour data (i.e., the fixed averaging time). The multiresolution basis utilized is a wavelet set, as it is the only wavelet set that satisfies Reynolds averaging (Howell and Mahrt, 1997; Mallat, 1998;



Vicers and Mahrt, 2003, 2006). In this study, we adopt the Hilbert-Huang transform (Huang et al., 1998, 1999), which is fully data-driven and local in both the physical and frequency domains (Huang et al., 1998; Flandrin et al., 2004). The advantages of researching turbulence data that are nonstationary and nonlinear have been proven previously (Huang et al., 2008; Schmitt et al., 2009; Huang et al., 2013; Wei et al., 2013, 2016, 2017).

The spectral gap was examined by studying the second-order Hilbert spectra (u′, v′, w′, θ′ and q′) for each 30-min period. The calculation method for the second-order Hilbert spectra comes from the arbitrary-order Hilbert spectral analysis (Huang et al., 2008), which is an extended version of the Hilbert-Huang transform. A joint probability density function (PDF), p(ω, A), can be extracted from $\omega_i$ and $A_i$ for all intrinsic mode functions. Then, the arbitrary-order Hilbert marginal spectrum is defined as an amplitude-frequency space by the marginal integration of the joint PDF p(ω, A)

$$\mathcal{L}_q(\omega) = \int p(\omega, A) A^q dA,$$   (4)

where ω represents the instantaneous frequency, A represents the instantaneous amplitude, $q \geq 0$ represents the arbitrary moment, and p(ω, A) represents the PDF. Specifically, h(ω) corresponds to the second-order Hilbert marginal spectrum($h(\omega) = \mathcal{L}_2(\omega) = \int p(\omega, A) A^2 dA$).

An automated algorithm was developed to objectively find the frequency location of the spectral gap. This method is based
on the characteristics in Fig. 4b in Wei et al. (2017). The spectral gap between turbulence and the submesoscale motions is identified as the frequency interval in which the second-order Hilbert spectral values are approximately constant, or the slope is approximately equal to 0. Figure 3 shows the second-order Hilbert spectra from the newly reconstructed data and raw data. As shown in Fig. 3a, the lower frequency limit of the spectral gap is ω = 0.008 Hz, and the upper frequency limit is not used to reconstruct the data. The part of the frequency larger than ω indicates the turbulence signal, and the part of the frequency
smaller than ω indicates nonstationary motion, which has a scale larger than that for turbulence. Then, we can search for the mode of the intrinsic mode function (IMF) (N=9 in this case), whose mean frequency is closest to ω. Therefore, modes 1-N are chosen to reconstruct the pure turbulence signal, and the remaining modes are chosen to reconstruct the signal of the submesoscale motion.

As shown in Figs. 3a−c, it is obvious that the reconstruction successfully eliminated the energy contained by large-scale
motion while retaining turbulent energy. The new spectrum, which is shown by the black dotted lines in Fig. 3, is consistent with the structure of the turbulent energy spectrum in the classic theory. We reconstructed the data for all time periods when the spectral gap existed in our dataset. The results of the total reconstructed data spectrum showed that the whole process removed the effects of large-scale motion and retained classic turbulence, while simultaneously obtaining large-scale motions.

**3.2 The local intermittent strength of turbulence**

We use the ratio of turbulent intensity compared to all signals to indicate the intermittent strength. To obtain the strength of submesoscale motion, we introduce the definition of the velocity scale for submesoscale motion from Mahrt (2007; 2009;



2010; 2011). The velocity scale for submesoscale motion represents the kinetic energy of submesoscale motions and is defined as:

$$V_{smeso} = \sqrt{u'^2_{smeso} + v'^2_{smeso} + w'^2_{smeso}} \,, \tag{5}$$

where $u'_{smeso}$, $v'_{smeso}$, and $w'_{smeso}$, represent the deviations reconstructed from the IMF corresponding to the submesoscale

5   motions during each 30-min period. Similarly, the turbulent velocity scale is defined as:

$$V_{turb} = \sqrt{u'^2_{turb} + v'^2_{turb} + w'^2_{turb}}, \tag{6}$$

where $u'_{turb}$, $v'_{turb}$, and $w'_{turb}$, represent the deviations reconstructed from the IMF corresponding to the turbulent motions during each 30-min period. Then, we can define the local intermittent strength from an energy point of view as

$$LIST = \frac{V_{turb}}{\sqrt{V^2_{smeso}+V^2_{turb}}}, \tag{7}$$

10   The value of the LIST is equal to the ratio of turbulent intensity in the acquired signal. When the LIST is large and close to 1, there are more turbulent components in the acquired signal, the influence of the submesoscale motion is weaker, and the intermittency is weaker. When the LIST is small and far from 1, there are fewer turbulent components in the collected signal, the influence of submesoscale motion is stronger, and the intermittency is stronger.




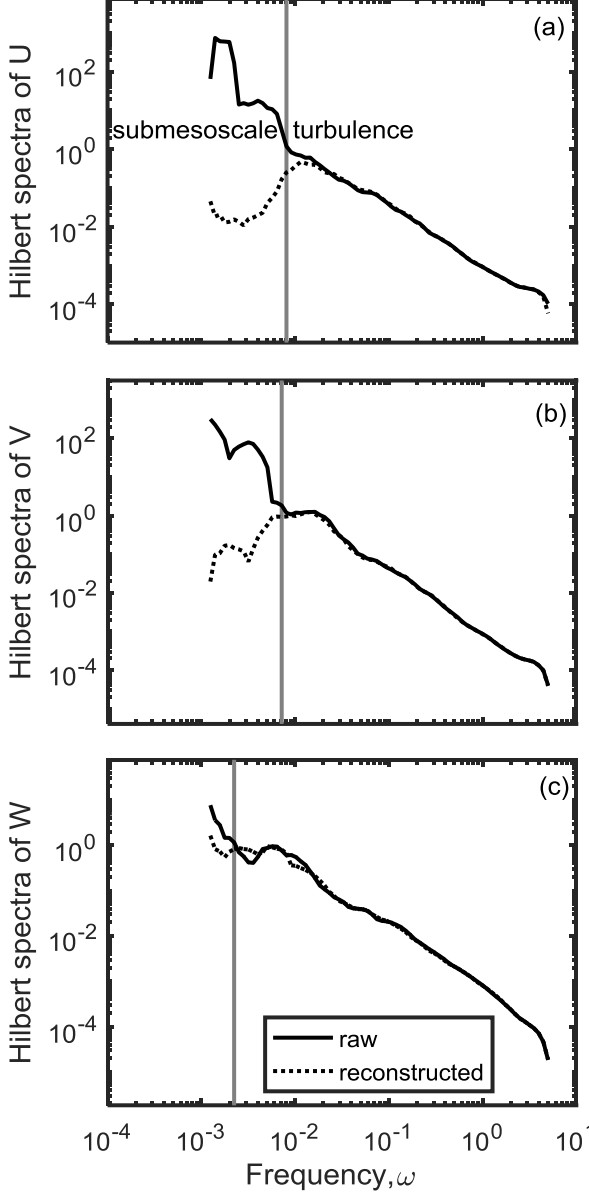

**Figure 3: Second-order Hilbert spectra of three wind speed components U (a), V(b) and W (c) at 08:00 on 31 December 2016 at the suburban site. The black solid line indicates the spectra from the raw data, and the black dotted line indicates the spectra from the reconstructed data for pure turbulence. The solid gray lines indicate the position of the spectral gap.**



## 4 Results and discussion

### 4.1 Comparison of the macro-statistical parameters for reconstructed turbulence and the results of the fixed averaging time

We use the automated algorithm to identify the spectral gaps in 1152 groups of 10-Hz high-frequency data from 16
December 2016 to 8 January 2017 at the urban and suburban sites. There are 440 groups of data in the along-wind direction,
519 groups of data in the cross-wind direction, 390 groups of data in the vertical direction, 467 groups of data in the scalar $\theta$,
and 501 groups of data in the scalar q, which is where the spectral gap occurs at the suburban site. The datasets
encompassing the spectral gap account for 38%, 45%, 34%,41%, and 43% of the total data at the suburban site, respectively.
The proportions are 41% (481), 37% (425), 35% (402), 41% (476), and 45% (520) for the u, v, w, t, and q components at the
urban site, respectively. The occurrence of a spectral gap is common throughout the pollution process, which means that the
effects of nonstationary motion on the collected signals are common throughout the entire process. Under these conditions,
the usual half-hour length is not suitable for stationary conditions. The eddy-correlation flux calculated using a conventional
averaging time of 30 min to define the perturbations is severely contaminated by poorly sampled mesoscale motions.

During the half-hour period, we refer to the variance in the pure turbulent fluctuations as the new results, and that calculated
via the classic EC system is referred to as the old variance results. A comparison of the TKE and variance parameters
($\sigma_u, \sigma_v, \sigma_w, \sigma_\theta$ and $\sigma_q$) between the new results and old results is shown in Fig. 4, and a comparison among the vertical heat
flux ($\overline{w'\theta'}$), vertical water-vapor flux ($\overline{w'q'}$), and momentum flux ($-\overline{u'w'}$) is presented in Fig. 5; the fitted lines are given in
both figures. When there is no spectral gap, the 30-min time length is suitable for stationary conditions; therefore, the classic
EC method is credible. Figure 4 and Fig. 5 present only the results when there is a spectral gap (i.e., when there is
nonstationarity at the suburban site).



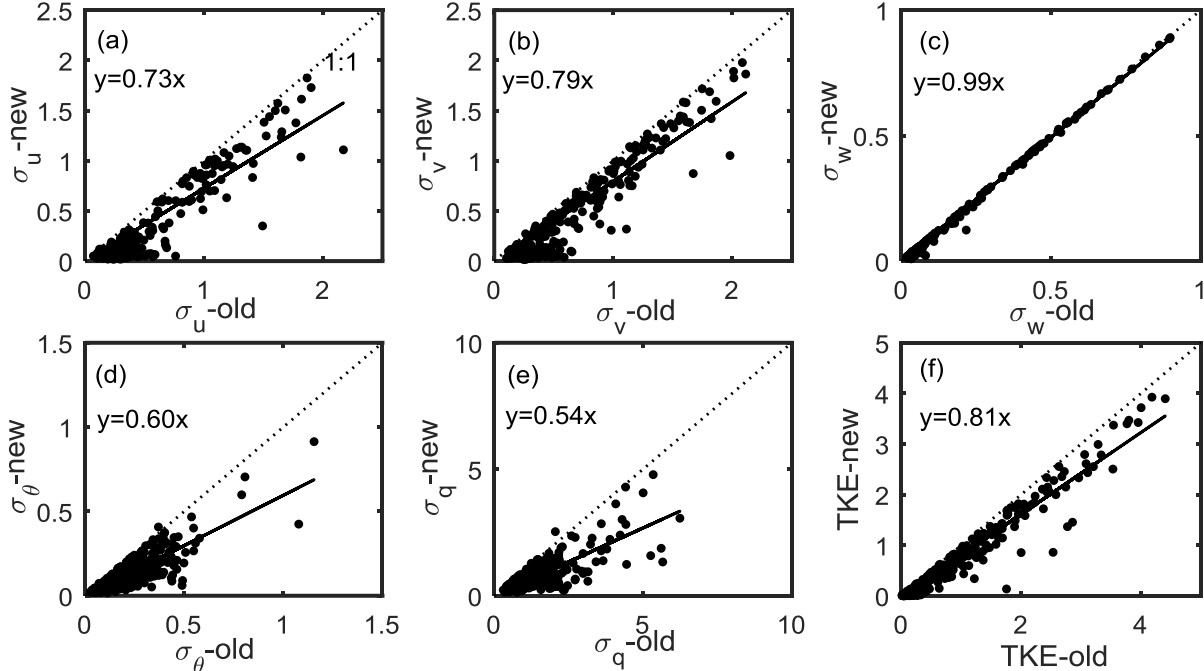

**Figure 4: Comparison of $\sigma_u$ (a), $\sigma_v$ ( b), $\sigma_w$ (c), $\sigma_\theta$ (d), $\sigma_q$ (e) and TKE (f) from the new half-hour results with those from the old results from 16 December 2016 to 8 January 2017 at the suburban site. The black dotted line represents the 1:1 line in the figures. The black solid line represents the fitted results.**

All of the fitted results show that there are certain degrees of overestimated variance in Fig. 4. The slope of the fitted line is 0.81 for the TKE, as Fig. 4f shows, which means that the TKE originally calculated by the conventional method is overestimated by approximately 19%. The comparison of $\sigma_u$ and $\sigma_v$ shows the same pattern as that for the TKE. The traditional method for the half-hour eddy correlation overestimates by approximately 27% (slope of 0.73) and 21% (slope of 0.79) for $\sigma_u$ and $\sigma_v$, respectively. The overestimation of $\sigma_w$ is not as obvious as those for $\sigma_u$ and $\sigma_v$. The slope of the fitted

line in Fig. 4 c is 0.99, which indicates that the appearance of the spectral gap has fewer effects on the variance in vertical velocity. The overestimation is even more pronounced when the TKE and $\sigma_u$, $\sigma_v$ and $\sigma_w$ are small, which indicates that the overestimation can be more significant when the turbulence is weak. Similarly, we examine the variance in potential temperature and moisture content in Figs. 4d and 4e, respectively. The variations in potential temperature and moisture content are also overestimated; the difference is that the overestimation percentages are larger. The slopes of the fitted lines

in Figs. 4d and 4e are 0.60 and 0.54, respectively, which means that the traditional method for the half-hour eddy correlation overestimates approximately 40% and 46% of $\sigma_\theta$ and $\sigma_q$, respectively, when the spectral gap occurs. Such a substantial overestimation cannot be ignored. In general, the variations in $\sigma_u$, $\sigma_v$, $\sigma_w$, $\sigma_\theta$ and $\sigma_q$ calculated by the traditional EC method for 30 min are all overestimated, especially when the turbulence is weak, due to the impact of submesoscale motion. In



addition, the overestimation in the horizontal direction is more significant than that in the vertical direction. However, the overestimation of these quantities cannot be ignored.

After separating pure turbulence and submesoscale motion from the signal, the pure turbulent flux can be calculated during heavy haze pollution periods. The comparison of the pure turbulent flux with the flux calculated by the eddy-correlation

method at a fixed averaging time is presented in Fig. 5. The momentum flux, $-\overline{u'w'}$, is overestimated by approximately 13%, and the slope of the fitted line in Fig. 5c is approximately equal to 0.87. The effect of nonstationary motion on heat transfer and water-vapor transfer is similar to that on momentum, as seen in Fig. 5. Overestimation can reach 12% (slope of 0.88) and 15% (slope of 0.85) for $\overline{w'\theta'}$ and $\overline{w'q'}$ in Figs. 5a and b, respectively. The reconstructed turbulence data results at the urban site show that the traditional half-hour eddy-correlation results have a certain degree of overestimation regarding the

turbulent fluxes and variances during the pollution period. However, the degree of overestimation in urban areas is less than that in suburban areas. A comparison of the macrostatistical parameters for turbulence from the new half-hour results with those from the old results at the city site is not shown here.

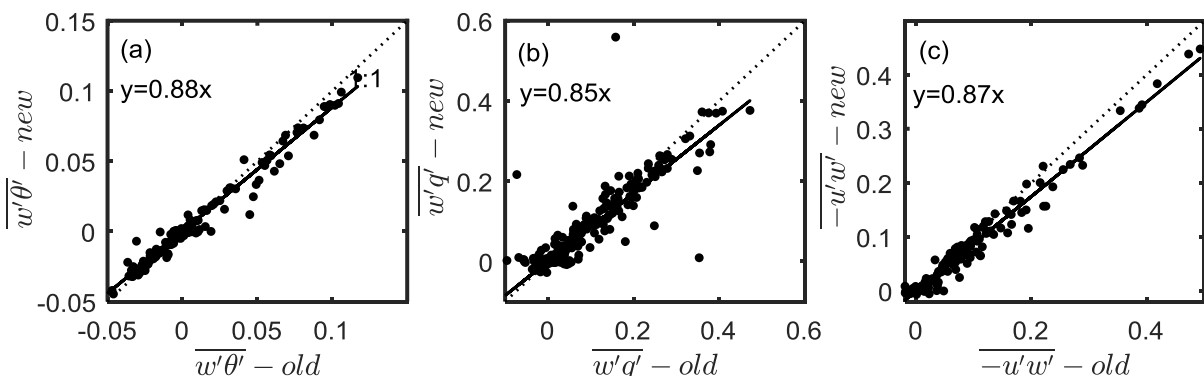

**Figure 5: A comparison of the vertical heat flux ($\overline{w'\theta'}$) (a), vertical water-vapor flux ($\overline{w'q'}$) (b) and momentum flux ($-\overline{u'w'}$) (c)**

**from the new half-hour results with those from the old results from 6 December 2016 to 8 January 2017 at the suburban site. The black dotted line represents the 1:1 line in the figure. The black solid line represents the fitted results**

In the majority of recent research, the EC technique has become the most frequently used method to estimate the flux between land surfaces and the atmosphere (Baldocchi et al., 2003). Under weak stability conditions, eddy-correlation fluxes calculated using a conventional averaging time of 5 min or longer to define the perturbations are severely contaminated by

poorly sampled mesoscale motions (Vickers and Mahrt, 2005). The average block time that has always been used is 30 min, which has been used in practice since Kaimal and Finnigan (1994), who proposed that a 30-min averaging time was a reasonable compromise for daytime analyses. From the above analysis, we can see that turbulent fluxes are overestimated due to the effects of submesoscale motions when there is a spectral gap. During the heavy haze pollution process, turbulence is weak, which causes this overestimation to become even more apparent. Zhong et al. (2017, 2018) proposed that heavy

pollution events in Beijing are characterized by the transport stage (TS), whose formation of aerosol pollution is primarily





caused by pollutants transported from regions south of Beijing, and the cumulative stage (CS), where the cumulative explosive growth in PM$_{2.5}$ mass concentration is dominated by stable stratification characteristics in the atmosphere, such as slight or calm southerlies, anomalous inversions near the ground, and moisture accumulation. According to the classification criteria for the CS and TS, we briefly compared the characteristics of turbulent fluctuation, variance and flux at different

5    times during the pollution process from 15 December to 23 December 2016, as shown in Fig. 6. Due to the consistent patterns of different variables, only the results for horizontal velocity are shown in this paper. We can clearly see from Fig. 6 that the turbulent fluctuation, variance, heat flux and momentum flux are obviously smaller during polluted periods (CS and TS) than those during clean periods. Therefore, the exchange of fluxes between the ground and atmosphere can be easily overestimated with the traditional eddy-correlation method during the pollution process, which can result in false forecasts of

10   contaminant concentrations and pollution levels. Some works have found that current pollution forecasts tend to underestimate pollutant concentrations (Li et al., 2016). The overestimation of turbulent flux during heavy pollution episodes may be one of the reasons for this result.

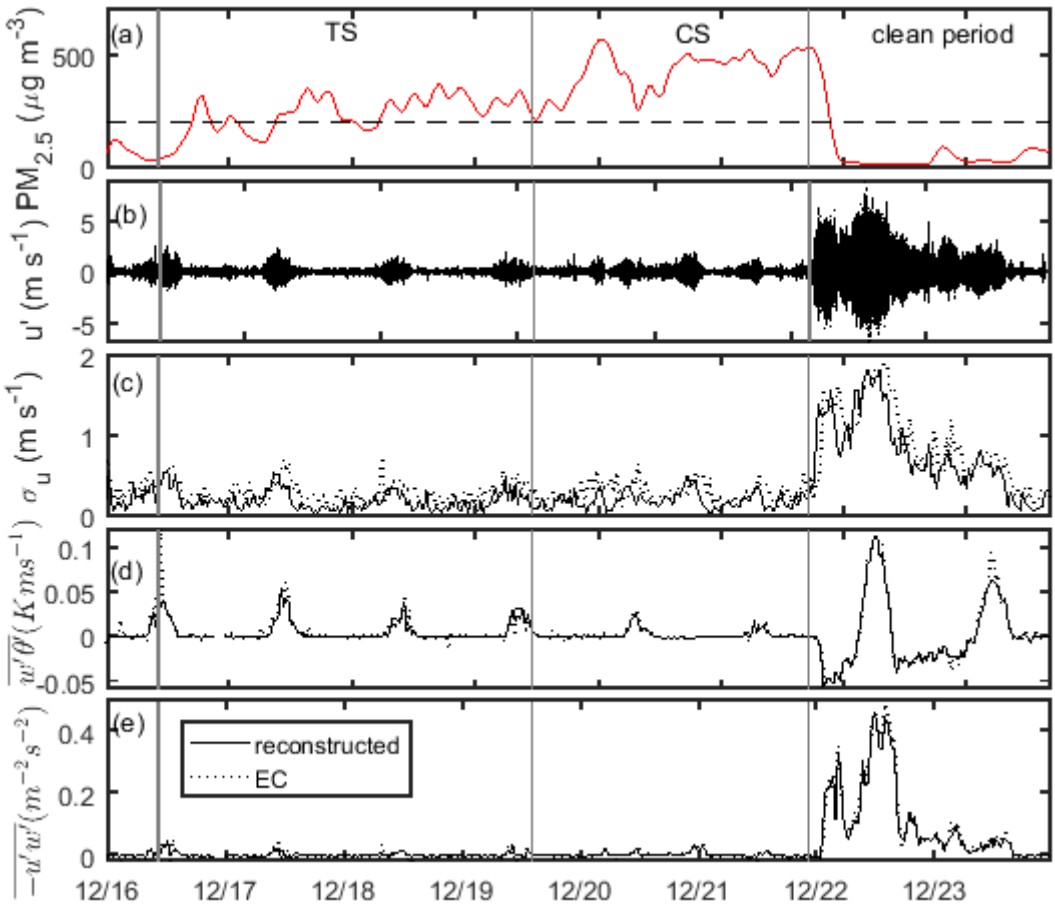





**Figure 6: Time series of (a) PM$_{2.5}$ concentration, (b) horizontal wind speed, (c) variation in horizontal wind speed, (d) vertical heat flux ($\overline{w'\theta'}$), and (e) momentum flux ($-\overline{u'w'}$) from 15 December to 23 December 2016 at the suburban site.**

## 4.2 The relationship between the local intermittent strength and pollution

After reconstruction via the automated algorithm, pure turbulent data can be obtained. The LIST from 16 December 2016 to
8 January 2017 is discussed in this section. We calculated the LIST of the urban and suburban sites separately and discussed
the relationship between the LIST and degree of pollution.

Figure 7 shows the time series for (a) PM$_{2.5}$ concentrations, (b) the intermittency factor (IF) of vertical velocity, (c) the
velocity scale for submesoscale motions ($V_{smeso}$), (d) the velocity scale for turbulence ($V_{turb}$), and (e) the LIST for
turbulence from 16 December 2016 to 8 January 2017 at the suburban site. Figure 8 shows the results of the same variables
at the city site. Wei et al. (2018) has shown that there is a relationship between the IF, which characterizes the intensity of
intermittent mixing, and the concentration of PM$_{2.5}$. In this section, we also present the IF of the vertical wind speed in Fig.
7b and Fig. 8b to discuss internal relations and differences with the local intermittent strength. Low PM$_{2.5}$ concentrations
correspond to large values of IF, $V_{smeso}$, and $V_{turb}$ and weak intermittent intensity (i.e., the LIST is closer to 1) in Figs. 7 and
8. The decrease in PM$_{2.5}$ concentration corresponds to the increases in IF, $V_{smeso}$, and $V_{turb}$ and the decrease in local
intermittent intensity. For several heavy pollution events with PM$_{2.5}$ concentrations higher than 200 μg m$^{-3}$, the absolute
value of the IF is smaller, the intensity of the submesoscale motion is weaker than that during the clean period, and the
energy of the turbulent flow is also weaker. Correspondingly, the local intermittent intensity is strong, as the LIST is farther
from 1. Specifically, for the period of explosive growth in pollutant concentration (CS) during the two pollution processes of
Case I and Case II in Fig. 7, the intermittent intensity (LIST) is significantly stronger (weaker) than that during the clean
period and other weak pollution events, with a concentration of approximately 200 μg m$^{-3}$. The two processes Case I and
Case II are similar in that both experience a rapid decline in the concentration of PM$_{2.5}$ (from a heavy pollution level greater
than 500 μg m$^{-3}$ to one less than 30 μg m$^{-3}$) on a time scale of several hours. The difference is that there is a clean period
approximately 1 day after the rapid decline in the Case I process, but in the Case II process, the PM$_{2.5}$ concentration
decreases sharply in just a few hours and then rapidly increases to the 600 μg m$^{-3}$ level. By comparing the rapid reduction in
the PM$_{2.5}$ concentrations during the Case I and Case II processes, we find that there is no significant difference in
submesoscale motion strength, whereas turbulent intensity during the Case I process is significantly greater than that during
the Case II process, which leads to a weaker (stronger) LIST during the Case I (Case II). During the Case II process,
although the LIST (intermittent intensity) increases (decreases) when the PM$_{2.5}$ concentration begins to decrease, the
larger(weaker) LIST (intermittent intensity) does not maintain but decreases (increases) rapidly, which indicates that the
pollutant concentration does not fully diffuse and rapidly increases again to 600 μg m$^{-3}$ within a few hours. The difference
between these two processes also indicates that the diffusion of pollutants directly depends on the diffusion of turbulent flow
during heavy pollution processes. Submesoscale motion may be an important source of turbulent energy under stable
weather conditions, but it is not directly involved in the dissipation of pollutants.



From the analysis of the clean and polluted periods, we can see that the submesoscale motion persists throughout the 30-min acquired signal. During heavily polluted periods, the intensity of turbulence is weak, and the effects of submesoscale motion on weak turbulence begin to appear. During clean periods, the intensity of turbulence is strong, and the effect of submesoscale motion is relatively small. Meanwhile, the submesoscale motion is a source of turbulent energy when the

turbulence is very weak under heavy pollution and stable weather conditions. It can be said that turbulence derives energy from nonstationary, large-scale motion and has a very large effect on the concentration of pollutants over time scales of hours (e.g., Case I and Case II). We also note the relationship between the LIST and IF. Wei et al. (2018) mentioned that the larger the absolute value of the IF is, the stronger the intermittent mixing. From the perspective of intermittent mixing strength, the IF tends to be consistent with the LIST defined in this paper. When the pollutant concentrations decreased

sharply, intermittent mixing increased, the IF absolute value increased, the LIST (intermittent intensity) increased(decreased), and turbulent exchange was enhanced.

The tendency of the LIST under pollution conditions in Fig. 8 is consistent with that in Fig. 7, which indicates that the above analysis conclusions apply not only to suburban sites but also to urban sites. However, there is a significant difference between the urban and suburban sites: the LIST (intermittent intensity) at suburban sites is weaker (stronger) than that at

urban sites. The value of $V_{smeso}$ at the city site is slightly smaller than that at the suburban site, and the value of $V_{turb}$ at the city site is larger than that at the suburban site. Correspondingly, the value of the LIST at the city site is significantly greater than that at the suburban site, which means that the dynamic effect of the underlying surface of the urban area is stronger than that of the flat underlying surface of the suburban area. Next, in Sections 4.3 and 4.4, we verify the results from the statistical characteristics of turbulence and the daily changes in turbulent flux during polluted and clear periods, respectively.





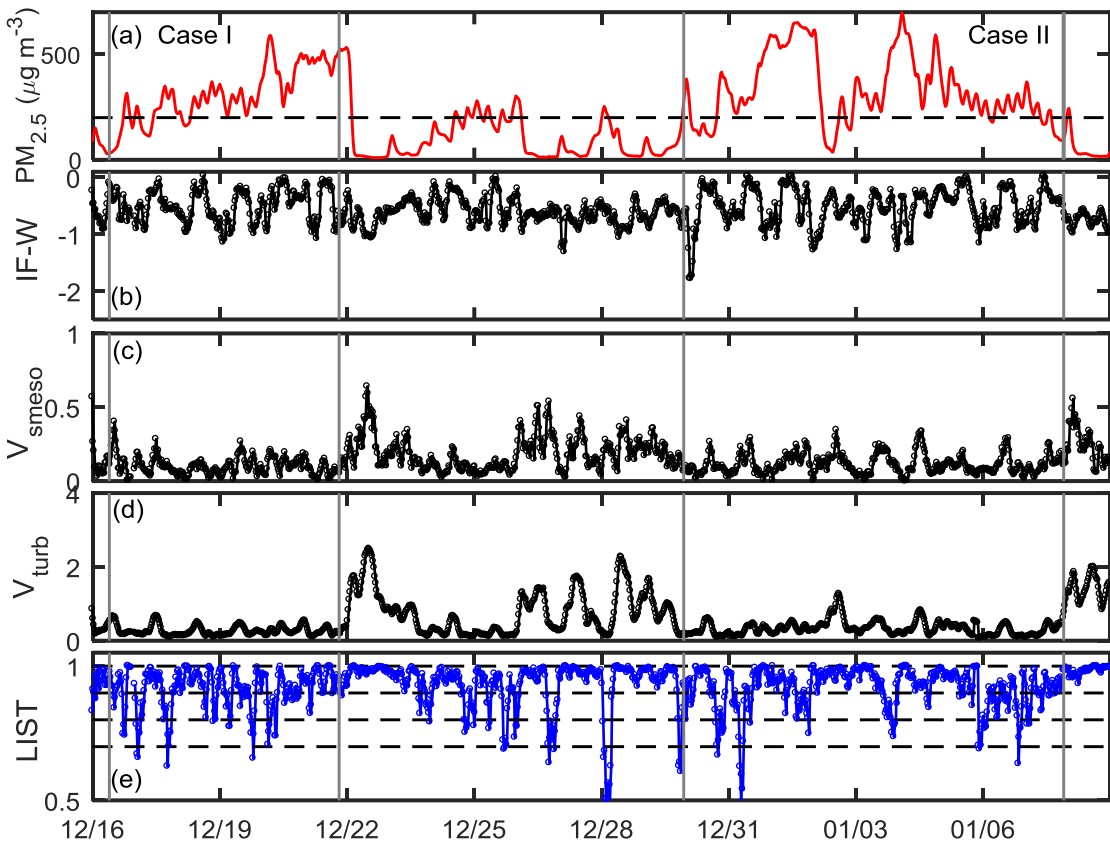

**Figure 7: Time series of the (a) PM₂.₅ concentrations, (b) intermittency factor (IF) of vertical velocity, (c) velocity scale of submesoscale motions ($V_{smeso}$), (d) velocity scale of turbulence ($V_{turb}$), and (e) LIST from 16 December 2016 to 8 January 2017 at the suburban site.**



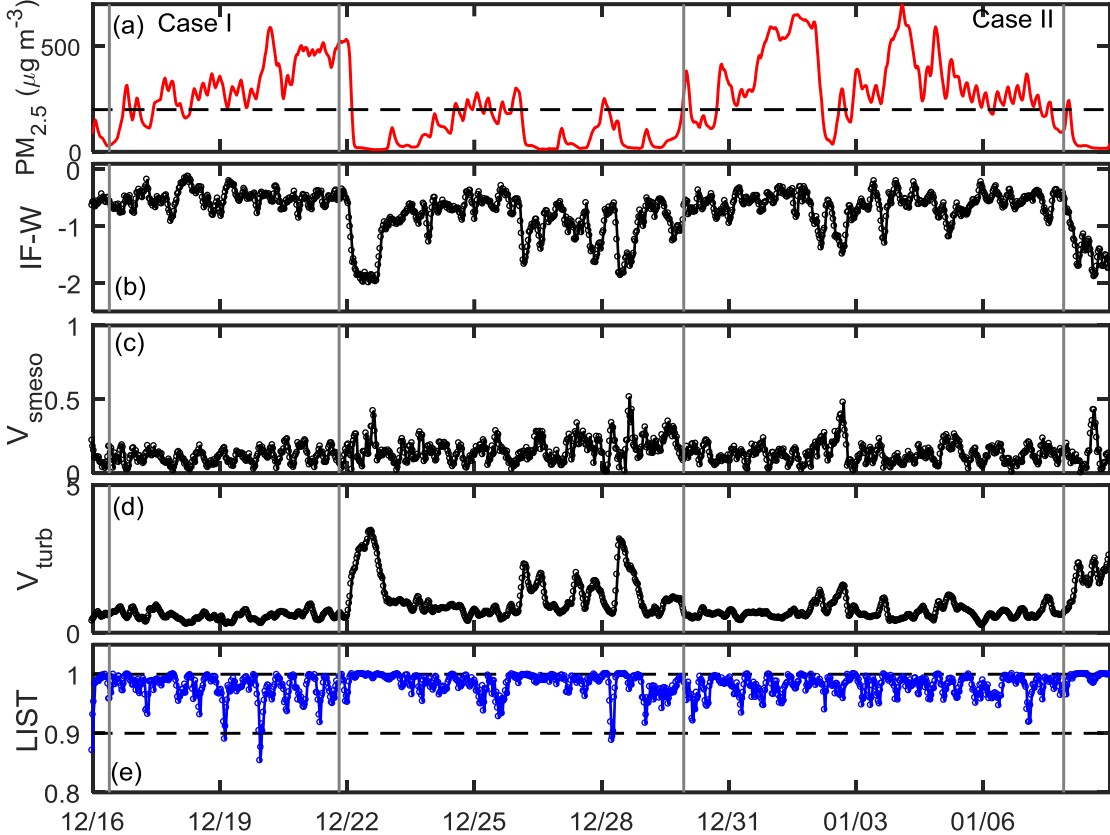

**Figure 8: Time series of the (a) PM₂.₅ concentrations, (b) intermittency factor (IF) of vertical velocity, (c) velocity scale of submesoscale motions ($V_{smeso}$), (d) velocity scale of turbulence ($V_{turb}$), and (e) LIST from 16 December 2016 to 8 January 2017 at the city site.**

### 4.3 Macrostatistical characteristics of turbulence

Figures 9 and 10 show the relationship of the normalized standard deviations in wind speed in the horizontal and vertical directions ($\sigma_u/u_*$, $\sigma_v/u_*$, and $\sigma_w/u_*$) with potential temperature ($\sigma_\theta/|\theta_*|$) and moisture content ($\sigma_q/|q_*|$) as functions of the stability parameter $\zeta$ at the suburban site, respectively. It should be noted that the reconstructions of the data have two benefits. One is that the discrete situation of the fitted line has been significantly improved after reconstruction. The other is that the statistical characteristics of turbulence are more consistent with the classic statistical pattern for turbulence, and the normalized standard deviations in potential temperature and water vapor under near-neutral conditions are closer to those of Panofsky et al. 1984 and Wyngaard (1971). The figures regarding the macrostatistical characteristics of turbulence before reconstruction are not shown here.



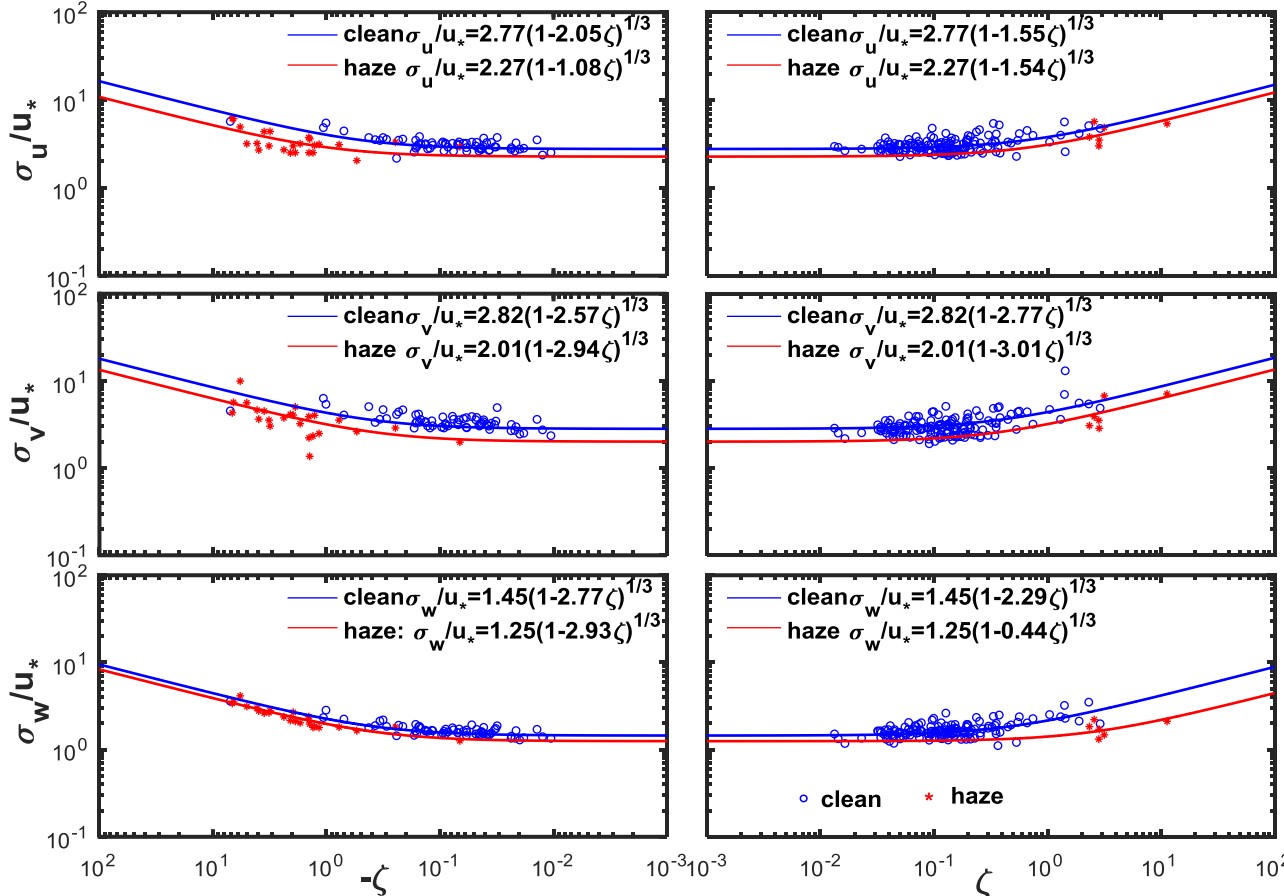

**Figure 9: Normalized standard deviations in wind speed in the horizontal and vertical directions ($\sigma_u/u_*$, $\sigma_v/u_*$, and $\sigma_w/u_*$) as functions of the stability parameter $\zeta$. The solid (dashed) line in the figure represents the results for polluted (clear) weather conditions. Observations marked with ∗ (°) were made under polluted (clear) weather conditions.**





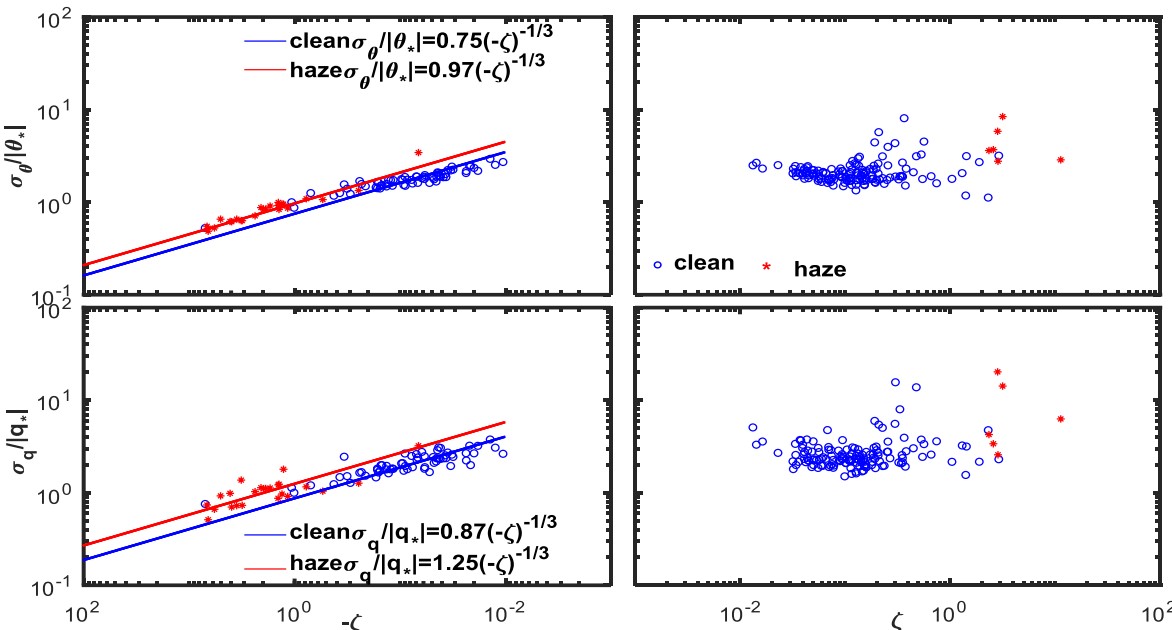

**Figure 10: Normalized standard deviations in the potential temperature ($\sigma_\theta/|\theta_*|$) and moisture content ($\sigma_q/|q_*|$) as functions of the stability parameter $\zeta$. The solid (dashed) line in the figure represents the results for polluted (clear) weather conditions. Observations marked with $*$ (°) were made under polluted (clear) weather conditions.**

Figure 9 shows that under both polluted and clear weather conditions, the relations of the normalized standard deviations in horizontal and vertical wind speeds with the stability parameter $\zeta$ follows the 1/3 power law well under stable and unstable stratification conditions. The normalized standard deviation was more dispersed in the horizontal direction than that in the vertical direction, which is consistent with the results presented in Zhang et al. (2004) and Ma et al. (2002) and shows that the physical characteristics of the land surface (e.g., topography and roughness) affect the statistical properties of the vertical

wind speed less than those of the horizontal wind speed. The normalized standard deviations in the horizontal and vertical wind speeds are nearly constant under near-neutral conditions (i.e., -0.01 < $\zeta$ < 0.01), and the constants are slightly larger than those from Panofsky et al. (1984), which is similar to the results from other suburban works (i.e., Zhang et al. (1991), Roth et al. (1993) and Su et al. (1994)). Figure 9 shows that the values during clean periods (circle) are slightly larger than those during haze periods (asterisk) under unstable conditions. Ren et al. 2018 (Fig. 4) shows the significant difference in the

normalized standard deviations in the horizontal and vertical wind speeds between clear and haze periods. The reconstructed data from the city site also show the same conclusion (the figures are not shown in this paper), which means that the dynamic effect of turbulence during pollution episodes is reduced significantly at both the city and suburban sites. At the same time, the normalized standard deviations in wind speed in three directions over the suburban area are smaller than those




over urban areas under near-neutral conditions, which shows that the dynamic effect of turbulence at the urban site is stronger than that at the suburban site.

Figure 10 shows the relationships of the normalized standard deviations in potential temperature and moisture with the stability parameter $\zeta$. Given unstable stratification, $\sigma_\theta/|\theta_*|$ and $\sigma_q/|q_*|$ fit the function $\zeta^{-1/3}$. The fitting coefficients are larger than the typical value of 0.95 presented in Wyngaard (1971), which used data from a grassland area in Kansas (United States). Similar to the conclusion for the normalized standard deviations in the three-directional wind speeds, Ren et al. 2018 (Fig. 5) shows the slight differences in the normalized standard deviations in potential temperature between clear and haze periods, while Fig. 10 in this paper shows a significant difference between clear and haze periods. Since the time periods studied are the same, the difference in the performance of these turbulence statistics during polluted and clean periods over city and suburban sites indicates the impact of urbanization. In other words, the pollution process has a more significant influence on the turbulent statistical characteristics of potential temperature and moisture in the suburbs and a less significant impact on those in the urban area.

### 4.4 The impact of urbanization on turbulent transport during heavy pollution episodes

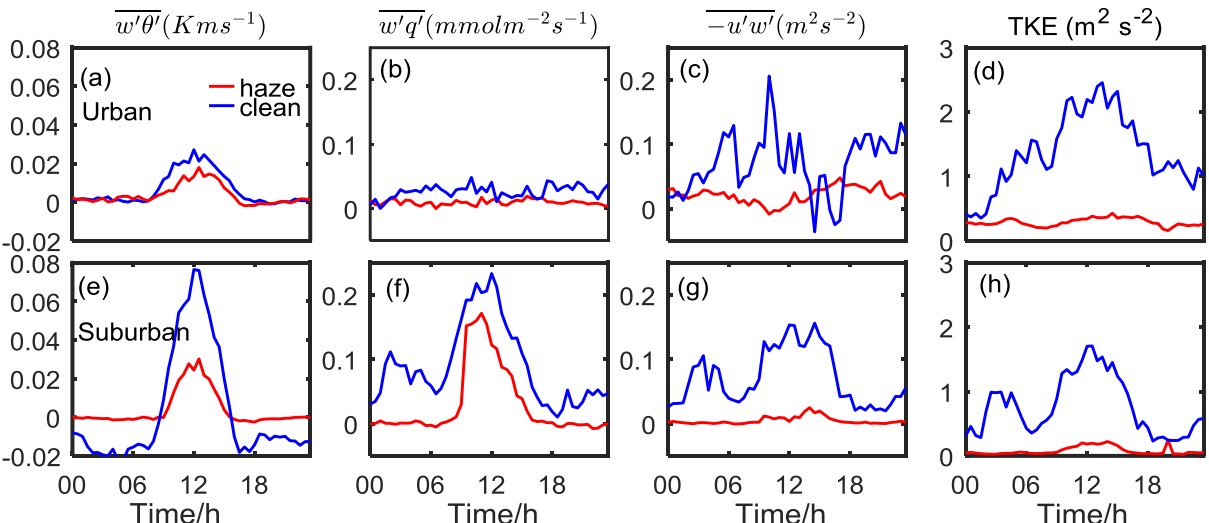

**Figure 11: Diurnal variations in the mean vertical heat flux ($\overline{w'\theta'}$) (a), vertical water-vapor flux ($\overline{w'q'}$) (b), momentum flux ($-\overline{u'w'}$) (c) and TKE (d) under polluted weather (solid line) and clear weather (dotted line) conditions over the urban site. Diurnal variations in these variables over the suburban site are shown in (e), (f), (g), and (h), respectively.**

Figure 11 shows the diurnal variations in the mean $\overline{w'\theta'}$, $\overline{w'q'}$, $-\overline{u'w'}$ and TKE under both polluted weather and clear weather conditions. The sensible heat flux (vertical heat flux) exhibits unimodal diurnal variations during both clean and polluted periods over either the suburban site or city site, as shown in Figs. 11a and e, respectively. During the clean period, the peak in the daily sensible heat flux at the suburban site appears around 12 noon (local time), reaching a maximum of 0.07





K m s$^{-1}$; the peak in the daily sensible heat flux at the urban site also appears around 12 noon, with a peak value close to 0.04 K m s$^{-1}$, which is less than that in the suburban area. One possible reason is that the buildings in the urban area are denser and have higher reflectivity, resulting in less net radiation and, consequently, a lower peak in the sensible heat flux. Another difference in the sensible heat flux between the city and suburban sites during the clean period is that the sensible heat flux

over the suburbs is negative at night, with a downward heat transfer, but not in urban areas. This pattern may be due to the emission of anthropogenic heat sources at nighttime in cities and the difficulty of dissipating heat over tall and dense buildings; therefore, there is not a vast temperature difference between the surface of the city and the upper air and, accordingly, the sensible heat flux is not completely below zero. Previous research on city and suburban sites in Helsinki has also found the same features (Nordbo et al., 2013). During the pollution period, the peak in sensible heat flux at the urban

site dropped to approximately 0.02 K m s$^{-1}$. In addition, other trends were exactly the same as those during the clean period. At the same time, the peak in sensible heat flux at the suburban site dropped to 0.02 K m s$^{-1}$, and the daily change in sensible heat flux was somewhat different than that during the clean period (i.e., the characteristic of downward sensible heat flux at night disappeared completely). In general, pollution caused a reduction in the upward transport of sensible heat flux, and it had a weaker impact on the city and a stronger impact on the suburbs. In addition, pollution can also hinder the downward

transmission of nighttime sensible heat flux in the suburbs.

The daily changes in latent heat flux over the urban and suburban areas are completely different. During the clean period, the latent heat flux at the city site (Fig. 11b, dotted line) does not have obvious daily variations and fluctuates at approximately 0, which is related to low winter temperatures, the presence of water in the form of solid ice, and very dry air. However, the daily changes in latent heat in the suburbs during the clean period show a unimodal change, reaching a peak at

approximately 12 noon. It seems that the pollution process has no significant effect on the transport of latent heat over the urban area. As the solid line in Fig. 11b shows, the latent heat flux over the urban area during the pollution period remains low, fluctuating at approximately 0. However, the pollution process has a greater impact on the transfer of latent heat in the suburbs. As shown by the dotted line in Fig. 11f, the transfer of latent heat is reduced throughout the day.

The magnitude of the momentum flux is related to the roughness of the underlying surface, wind speed, and stability of the

boundary layer. During the clean period, the momentum flux over the city site experiences a wide range of changes, and the peak value is larger due to the high roughness of the city's underlying surface; the suburban momentum flux also has obvious daily changes, but the peak value is slightly lower than that over the city site. During the pollution period, the momentum fluxes at the urban and suburban sites are significantly lower than those during the clean period and remain low throughout the day. However, at this time, the momentum flux over the city is greater than that over the suburbs, as Figs. 11c and 11g

show, respectively. The change in TKE is similar to that in momentum flux. During the clean period, the TKE values over the urban and suburban sites all showed obvious unimodal diurnal variations. The TKE values over the urban area were greater than those over the suburban area, which is due to the strong turbulence in wake vortices caused by underlying surfaces, such as high buildings in urban areas. During the pollution process, the urban and suburban TKEs both decreased



significantly and remained low throughout the day. However, the TKE over the city site was still greater than that over the suburban site.

Based on the above analysis, we know that the sensible heat flux and latent heat flux of the urban site are less than those of the suburban site, and the momentum transport and TKE of the urban areas are greater than those of the suburban sites due to

different underlying surfaces. The sensible heat flux, latent heat flux, momentum flux, and TKE in the urban and suburban areas are all affected when pollution occurs. Pollution inhibits the material and energy exchanges between the surface and atmosphere. Moreover, the impact of the pollution process on the suburbs is much greater than that on the urban area. If the turbulent structure over the flat surface of the suburban area is considered the normal state for turbulent structures in this area during the studied period, then the urban turbulent structure is influenced by urbanization. From this perspective,

urbanization has reduced the impact of pollution on water and heat exchanges between the surface and atmosphere. In conjunction with Section 4.2, the value of the LIST in urban areas is greater than that in suburban areas, which means that the intermittent intensity of urban turbulence is less than that of suburban turbulence. Therefore, the turbulence signal over the urban site is stronger than that over the suburban site, and pollution in urban areas is weaker than in adjacent suburbs.

## 5 Conclusions and discussions

In this paper, we developed an automated algorithm to identify the spectral gap to separate pure turbulence and submesoscale motions from a 30-min signal based on the arbitrary-order Hilbert spectral method. We used this automated algorithm to analyze turbulence data observed from several severe haze pollution episodes in Beijing and its nearby suburbs from 16 December 2016 to 8 January 2017. The datasets with a spectral gap accounted for approximately 30% of the total data, indicating that the eddy-correlation flux calculated using a conventional averaging time of 30 min to define

perturbations is severely contaminated by poorly sampled mesoscale motions. Due to space limitations, only the detailed suburban site results are listed in the text. The results of the urban site reflect are consistent with the conclusions of the suburban site. A comparison between the reconstructed variances of $u'$, $v'$, $w'$, $\theta'$ and $q'$ and those calculated by the conventional method revealed overestimations of approximately 27%, 21%, 1%, 40%, and 46%, respectively, when there was a spectral gap. The vertical wind speed was overestimated less than the horizontal wind speed. The scalar, potential

temperatures were overestimated more than the vector, wind speed. The calculations of the fluxes were also overestimated. The momentum flux, heat flux and water-vapor flux were overestimated by approximately 13%, 12%, and 15%, respectively. We can see from these comparisons that the overestimation of flux during haze events cannot be ignored.

After reconstruction via the automated algorithm, pure turbulent data can be obtained. Then, we explore the relationship between the LIST and pollution. The results indicate that when pollution is heavy, the LIST is smaller, and the intermittency

is stronger; when pollution is lighter, the LIST is larger, and the intermittency is weaker. During the heavy pollution process, air quality is determined by turbulent motion on the hourly scale. At the same time, the LIST at the city site is greater than that at the suburban site, which means that the intermittency over the complex city surface is weaker than that over the flat





terrain of the suburbs. Urbanization seems to reduce the intermittency during heavy haze pollution episodes. The results were validated via the statistics on the impact of urbanization on turbulence and turbulent transport during polluted and clean periods.

The results for the statistical characteristics of turbulence show a significant difference in the normalized standard deviations in the horizontal and vertical wind speeds between clear and haze periods at the suburban site, which is the same as that at the city site; significant differences in the normalized standard deviations in potential temperature and water-vapor content between clear and haze periods are also observed at the suburban site, which are substantially different from those at the city site. These conclusions are consistent with the diurnal variations in mean flux under both polluted weather and clear weather conditions at the city site and suburban site. The sensible heat flux, latent heat flux, momentum flux, and TKE over the urban and suburban areas are all affected when pollution occurs. Pollution inhibits material and energy exchanges between the surface and atmosphere. Moreover, the impact of the pollution process on suburban areas is much greater than that on urban areas. Urbanization seems to help reduce the consequences of pollution.

*Data availability.* Data used in this study are available from the corresponding author upon request (hsdq@pku.edu.cn).

*Competing interests.* The authors declare that they have no conflict of interest.

*Acknowledgement.* This work was jointly funded by grant from National Key R&D Program of China (2016YFC0203300), the National Natural Science Foundation of China (91544216, 41705003, 41675018, 41475007).

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
