# Peer review of "Effects of turbulence structure and urbanization on the heavy haze pollution process"

_Atmospheric Chemistry and Physics, 2018_

## Referee Comment (RC1) · Anonymous Referee #1 · 22 Oct 2018

General comments: This manuscript analyzed the turbulence data observed from several severe haze pollution episodes in Beijing and its nearby suburbs by using the developed automated algorithm of identifying the spectral gap to separate pure turbulence and submesoscale motions from a 30-min signal based on the arbitrary-order Hilbert spectral method. Although I agree that the motivation of this study is good and its results are interesting, the presented study still needs some minor revision including the improvement in English before consideration for publication.

Specific comments: 1.In the abstract, "Urbanization seems to help reduce the consequences of pollution" may be somewhat misleading. 2.Line 16 in Page 3, "turbulence data observed from several severe haze pollution episodes", "from" should be changed to "in" or "during". 3.The data source of PM2.5 mass concentration, horizontal wind

[Figure]
Interactive
comment

speed, virtual temperature and water vapor mixing ratio need be described. The details of all the data used in this study should be included in the Section. 4.It may be better to modify the title of Section 3 as "Methodology" or "Methodology of reconstructing signals". Section 2.2 may be merged into "Methodology". In a word, Section 2 and Section 3 should be rearranged. 5.In Figure4, the comparison is made between the new half-hour results with those from the old results. Which is the reference? Is the overestimation of the variations in the variables calculated by the traditional EC method for 30 min referenced to the results using the new method? Then, what is the reference to assess the new method? Here, the description is confused. 6.What's the difference of PM2.5 in Fig.6-8 and Fig.2? How about the difference of data source? More details need be described. 7.In Fig. 9-11, the description of lines is wrong.

---

## Referee Comment (RC2) · Anonymous Referee #2 · 14 Nov 2018

The authors investigated the influence of turbulence structure and urbanization on the heavy haze pollution process, and did a very detailed analysis. The results provide valuable information on the interaction between pollution and turbulence flux and the differences between urban and rural area during heavy haze time. However, further study according to following comments is needed:

Major comments:

1) the authors mentioned "the impact of the pollution process on suburb areas is much greater than that on urban areas. Urbanization seems to help reduce the consequences of pollution" in the end of abstract. What are the dynamic/physical mechanism behind that? What is the difference in PM2.5 source at urban areas and suburb areas?

2) The heights of two observational sites are different, one at 25m and the other at 10m. What are differences of temperature, wind speed, specific humidity and PM2.5 caused by the height difference?

The authors used one point to represent the whole Beijing, how much is the difference in surface temperature, wind speed, specific humidity and PM2.5 at different locations of urban Beijing?

3) The authors pointed out the differences between clear and pollution days in the mass and energy exchanges at surfac e and concluded that the pollution inhibits the mass and energy exchange at surface. However, weak vertical mixing and surface wind can cause pollutants to accumulate and lead to heavy pollution.

Minor comments:

1)Figure 2a only has one line, but 2b and 2c have two lines. Figure 2a should have two line to show the results from two stations.

---

## Author Comment (AC1) · 17 Dec 2018

Dear reviewer,

 We all appreciate your hard work on this paper. These constructive opinions help to improve our work to a great extent. We did our best to respond to each comment and make this work well-organized. With the help of your detailed comments, some mistakes in the original manuscript were found and revised. Details are listed as follows:

General comments:
This manuscript analyzed the turbulence data observed from several severe haze pollution episodes in Beijing and its nearby suburbs by using the developed automated algorithm of identifying the spectral gap to separate pure turbulence and submesoscale motions from a 30-min signal based on the arbitrary-order Hilbert spectral method. Although I agree that the motivation of this study is good and its results are interesting, the presented study still needs some minor revision including the improvement in English before consideration for publication.

Specific comments:
1. In the abstract, "Urbanization seems to help reduce the consequences of pollution" may be somewhat misleading.

**Response:** Thanks for your comments. It was really somewhat misleading. We changed the expression to "The turbulent effects caused by urbanization seems to help reduce the consequences of pollution within the same weather and pollution source condition, because the turbulence intermittency is weaker and the reduction of turbulence exchange is smaller over urban underlying surface." in the abstract. Next, I will explain in detail why we made such changes.

Suburban site is located at locally flat underlying surface, as shown in Fig.1, is 20 km away from the urban site. At this distance, we believe that the impact of the large-scale weather situation on both sites is consistent. Recent studies have shown that Beijing's winter pollution process can be divided into pollutant transport stage and accumulation stage (Zheng et al., 2015a; Liu et al., 2013; Guo et al., 2014; Zhong et al. 2017; Zhong et al., 2018). Some studies have shown that emissions from outside Beijing can contribute 28–70% of the ambient $PM_{2.5}$ concentrations in Beijing (An et al., 2007; Streets et al., 2007; Wang et al., 2014; Chang et al. 2018). During the period we are concerned, as shown in Fig. 2, Zhong et al.'s research indicates that the pollution process is composed of the transportation of pollutants by southerly wind and accumulation of pollutants under the static wind (wind speed less than 2 m/s). Pollutant transport is dominated by large-scale weather processes, so the contribution to the source of $PM_{2.5}$ due to transport can be considered to be substantially consistent between the two sites separated by 20 km. In the accumulation stage of pollutants, it can be seen that the wind speed is less than 2 m/s in the layer below 1 km, and there is no longer large-scale transportation. At this time, the contribution of the local source may be highlighted. Due to the regulatory measures such as factory shutdowns implemented by the government in the fall of 2016 ("Bulletin on the State of China's Ecological Environment in 2016"), there were few major industrial sources in Beijing during our research phase, and

there are only a small number of residential sources. In fact, in the government regulation in 2017, the residential sources had also been strictly controlled. Although the suburban site is located in a relatively flat farmland, it is still in the vicinity of the Changping county. Therefore, the residential sources between the two sites may be different, but this difference is not significant compared to the large number of sources in the transportation process. In summary, we think that the sources of $PM_{2.5}$ of the two sites are generally consistent.

So, what we concluded in the manuscript is based on the background that the urban and suburban site are under the same weather and pollution source condition. That is why we changed the expression in that way.

[Figure]

**Figure 1** Google earth map of the suburban site within a range of approximately 1 km.

[Figure]

**Figure 2** Temporal variations in PM$_{2.5}$, PLAM, and vertical distributions of meteorological factors from 1 December 2016 to 10 January 2017 by Zhong et al. (2017), their Fig.1. (a) PM$_{2.5}$ mass concentration, (b) wind vector and wind velocity (shading; m s$^{-1}$), (c) temperature (shading; ℃), (d) RH (shading; %), and (e) PLAM. Red boxes correspond to original/transport explosive growth processes, while green boxes correspond to subsequent/cumulative explosive growth processes.

According to your comments, the following changes were made:

Changed the description in the abstract on the first page, line 25-28:

"The turbulent effects caused by urbanization seems to help reduce the consequences of pollution under the same weather and pollution source condition, because the turbulence intermittency is weaker and the reduction of turbulence exchange is smaller over urban underlying surface."

Figure 1 on page 6 in the revised manuscript has been modified to reflect the condition of the underlying surface around the two sites. The new Fig. 1 for the manuscript is shown as Fig. 3 here:

[Figure]

**Figure 3 Figure 1 in the manuscript**: Google Earth map of the observation sites in Beijing: (a) the observation site located in the urban underlying surface region (marked by the red pentagram) and the observation site located in the suburban underlying surface region with a flat terrain (marked by the red circle). The corresponding terrains (within a range of approximately 1 km) around the observation sites are shown in (b) and (c), respectively.

 Details of the suburban site are added on page 3, lines 29-30:
"Data over a locally flat underlying surface were collected at a continuous measurement site (40.16° N, 116.28° E) in the Beijing suburb. The observational site was set up in the middle of a vast and horizontal farmland, near the Changping county."
We added a description of the location relationship between the two sites on page 4, lines 20-32:
"Suburban site is 20 km away from the urban site. At this distance, the large-scale weather background is consistent. As flat terrain of the suburban site, it was used as a reference. The sources of $PM_{2.5}$ of the two sites are generally consistent."

2. Line 16 in Page 3, "turbulence data observed from several severe haze pollution episodes", "from" should be changed to "in" or "during".
**Response:** Thanks for pointing out that. "from" has been corrected to "during".

3. The data source of $PM_{2.5}$ mass concentration, horizontal wind speed, virtual temperature and water vapor mixing ratio need be described. The details of all the data used in this study should be included in the Section.
**Response:** Thanks for your suggestion. We have supplemented the details of the observation instruments and data of the two sites respectively.
We added some descriptions of the suburban site at page 4, lines 2-4. Details are

given as follows:

"The turbulence data such as the horizontal wind vector, virtual temperature and water vapor content were collected using a data logger (CR3000, Campbell Scientific, Inc., USA) at a frequency of 10 Hz and were averaged over an interval of 30 min for the analysis of meteorological elements."

We added some descriptions of the urban site at page 4, lines 11-17. Details are given as follows:

"The concentrations of $PM_{2.5}$ were collected using a Thermo-Fisher Sci. Co. instrument (series FH-62-C14), and 30-min averaging time series were performed to remove outliers. The system was equipped with an integrated $CO_2/H_2O$ open-path gas analyzer (LI-7500, LI-COR Biosciences, Inc., USA) and three-dimensional sonic anemometer-thermometer (IRGASON, Campbell Scientific, Inc., USA). The IRGASON was leveled and pointed north. The turbulence data such as the horizontal wind vector, virtual temperature and water vapor content were collected using a data logger (CR3000, Campbell Scientific, Inc., USA) at a frequency of 10 Hz and were averaged over an interval of 30 min for the analysis of meteorological elements."

We added a description of the location relationship between the two sites on page 4, lines 20-24:

"Suburban site is 20 km away from the urban site. At this distance, the large-scale weather background is consistent. As flat terrain of the suburban site, it was used as a reference. The sources of $PM_{2.5}$ of the two sites are generally consistent. The observations of $PM_{2.5}$ at urban sites are used to represent the evolution of the entire pollution process, as this study focuses on pollution processes rather than specific values. Since the observations at both sites are located in the surface layer, i.e. the constant flux layer, the values of turbulence flux are comparable."

4. It may be better to modify the title of Section 3 as "Methodology" or "Methodology of reconstructing signals". Section 2.2 may be merged into "Methodology". In a word, Section 2 and Section 3 should be rearranged.

**Response:** Thanks so much for your constructive advice. We modified the title of Section 3 as "Methodology". You are right that the description of method of calculating the turbulent quantities in Section 2.2 should be merged into Section 3. So we retain the content of data processing in Section 2.2 and move the content of description of method to the Section 3 which was rearranged.

The new Section 3.1 at page 8, lines 9-24 are given as follows:

"**3.1 Turbulent kinetic energy and turbulent fluxes**

The physical quantities used in this paper are turbulent kinetic energy (TKE), several variances ($\sigma_u$, $\sigma_v$, $\sigma_w$, $\sigma_\theta$ and $\sigma_q$), friction speed ($u_*$), and fluxes ($-\overline{u'w'}$, $\overline{w'\theta'}$ and $\overline{w'q'}$). Among these, the TKE is calculated as:

$$\text{e} = \frac{1}{2}\left(\overline{u'^2} + \overline{v'^2} + \overline{w'^2}\right), \tag{1}$$

the variance is calculated as

$$\sigma_u = \overline{u'u'},$$
$$\sigma_v = \overline{v'v'},$$
$$\sigma_w = \overline{w'w'}, \tag{2}$$
$$\sigma_\theta = \overline{\theta'\theta'},$$
$$\sigma_q = \overline{q'q'},$$

the turbulence flux is calculated as

$$\tau = \rho u_*^2 = \rho\overline{u'w'},$$

$$H = \rho C_p \overline{w'\theta'}, \tag{3}$$

$$E = \rho\overline{w'q'},$$

and the friction speed is calculated as:

$$u_* = \left[\left(-\overline{u'w'}\right)^2 + \left(-\overline{v'w'}\right)^2\right]^{\frac{1}{4}}. \tag{4}$$

"

The original Section 3.1 and Section 3.2 became the new Section 3.2 and Section 3.3

5. In Figure4, the comparison is made between the new half-hour results with those from the old results. Which is the reference? Is the overestimation of the variations in the variables calculated by the traditional EC method for 30 min referenced to the results using the new method? Then, what is the reference to assess the new method? Here, the description is confused.

**Response:** Thanks for your suggestion. Yes, the overestimation of the variations in the variables calculated by the traditional EC method for 30 min is referenced to the results using the new method. In this work, we recognize that the new method can get the pure turbulence part and eliminate the effects caused by sub-mesoscale motion. Because we can get the conclusion by comparison of the spectra between the raw data and reconstructed data. For example, Fig. 3 in the manuscript shows the second-order Hilbert spectra from the newly reconstructed data and raw data. The raw data spectrums, which are shown by the black solid lines in Fig. 3 in the manuscript, are inconsistent with the structure of the turbulent energy spectrum in the classic theory on frequency bands with lower frequency, that is, smaller than the frequency indicated by the grey solid lines. The new spectrum, which is shown by the black dotted lines in Fig. 3 in the manuscript, is consistent with the structure of the turbulent energy spectrum in the classic theory. It is obvious that the reconstruction successfully eliminated the energy contained by large-scale motion while retaining turbulent energy. Under the situation that there are spectra gaps, the turbulence data we obtained during the heavy pollution process contains the sub-mesoscale motion signal. Turbulent flux calculated by the traditional time-averaging method is contaminated by sub-mesoscale motions during the heavy pollution process. Similarly, this kind of contamination to turbulence flux caused by sub-mesoscale motion was also studied in some other works (Vickers and

Mahrt, 2006; Acevedo et al., 2006, 2007; Aubinet, 2008; Mahrt, 2010). All in all, we can find that the method we developed in this paper which is based on the Hilbert-Huang transform can get more realistic exchange between the surface and atmosphere during the heavy haze pollution.

However, you are right that the description here may easily cause confusion. We changed the expression of "old results" to "original results" in the manuscript which maybe can make it more clear. The figures involved the expression of "old" in the manuscript are Fig. 4 (on page 13) and Fig.5 (on page 15) which were also modified.

[Figure]

**Figure 4 in the manuscript**: Comparison of $\sigma_u$ (a), $\sigma_v$( b), $\sigma_w$ (c), $\sigma_\theta$ (d), $\sigma_q$ (e) and TKE (f) from the new half-hour results with those from the original results from 16 December 2016 to 8 January 2017 at the suburban site. The black dotted line represents the 1:1 line in the figures. The black solid line represents the fitted results.

[Figure]

**Figure 5 in the manuscript**: A comparison of the vertical heat flux ($\overline{w'\theta'}$) (a), vertical water-vapor flux ($\overline{w'q'}$) (b) and momentum flux ($-\overline{u'w'}$) (c) from the new half-hour results with those from the original results from 6 December 2016 to 8 January 2017 at the suburban site. The black dotted line represents the 1:1 line in the figure. The black solid line represents the fitted results.

6. What's the difference of PM$_{2.5}$ in Fig.6-8 and Fig.2? How about the difference of

**Response:** Thanks for your suggestion. The data source of $PM_{2.5}$ is the same in Fig.6-8 and Fig.2. We are sorry that we have not described clearly. In order to facilitate comparative analysis and display intuitively, the time series of $PM_{2.5}$ was added in Fig.6-8. The observations of $PM_{2.5}$ at urban sites are used to represent the evolution of the entire pollution process, as this manuscript focuses on pollution processes rather than specific values. As mentioned in the answer of the first question, the sources of $PM_{2.5}$ of the two sites are not much different. In fact, the trends in concentration of $PM_{2.5}$ across all environmental monitoring sites throughout the Beijing area are consistent, although there are some numerical differences. We choose three environmental monitoring stations, Changping (116.23°N, 40.22°E), Haidian (116.29°N, 39.99°E) and Daxing (116.40°N, 39.72°E), to prove that. The time series of mass concentration of $PM_{2.5}$ at the three environmental monitoring stations are shown in Fig. 6, their locations are shown in Fig. 7.

The data of $PM_{2.5}$ used in this manuscript is mainly to show the corresponding relationship between the trend of pollution development and intermittent turbulence, as shown in Fig.7 and Fig.8 in the manuscript. For the purposes of this work, the difference in the magnitude of the $PM_{2.5}$ values between the two sites does not affect the results. And because the pollution data from environmental monitoring sites have a large number of missing measurements during the study period, so we still use the observations from the urban site to represent the evolution of the entire pollution process.

An explanation of the data problem of $PM_{2.5}$ is added to Section 2, page 4, lines 21-22:

"The observations of $PM_{2.5}$ at urban sites are used to represent the evolution of the entire pollution process, as this study focuses on pollution processes rather than specific values."

[Figure]

**Figure 6** The time series of mass concentration of PM$_{2.5}$ at the three environmental monitoring stations in Beijing.

[Figure]

**Figure 7** Google Earth map of the observation sites in Beijing. The locations of the three environmental monitoring stations are marked by the red pentagram. The urban and suburban sites in manuscript are marked by red circle.

7.  In Fig. 9-11, the description of lines is wrong.

**Response:** Thanks for your suggestion. We are sorry for these faults. We corrected the descriptions of lines in Fig. 9-11 as follows:

"Figure 9: Normalized standard deviations in wind speed in the horizontal and vertical directions ($\sigma_u/u_*$, $\sigma_v/u_*$, and $\sigma_w/u_*$) as functions of the stability parameter $\zeta$. The red (blue) solid line in the figure represents the results under polluted (clear) weather conditions. Observations marked with $*$ ($°$) were made under polluted (clear) weather conditions." (page 18).

"Figure 10: Normalized standard deviations in the potential temperature ($\sigma_\theta/|\theta_*|$) and moisture content ($\sigma_q/|q_*|$) as functions of the stability parameter $\zeta$. The red (blue) solid line in the figure represents the results under polluted (clear) weather conditions. Observations marked with $*$ ($°$) were made under polluted (clear) weather conditions." (page 19, lines 2-4).

"Figure 11: Diurnal variations in the mean vertical heat flux ($\overline{w'\theta'}$) (a), vertical water-vapor flux ($\overline{w'q'}$) (b), momentum flux($-\overline{u'w'}$) (c) and TKE (d) under polluted weather ( red solid line) and clear weather (blue solid line) conditions over the urban site. Diurnal variations in these variables over the suburban site are shown in (e), (f), (g), and (h), respectively."

**References**

Acevedo, O.C., Moraes, O. L. L., Degrazia, G. A., and Medeiros, L. E.: Intermittency and the exchange of scalars in the nocturnal surface layer, Bound. Layer Meteor., 119, 41–55, 2006.

Acevedo, O. C., Moraes, O. L. L., Fitzjarrald, D. R., Sakai, R. K., and Mahrt, L.: Turbulent carbon exchange in very stable conditions, Bound. Layer Meteor., 125 (1), 49-61., 2007.

An, X., Zhu, T., Wang, Z., Li, C., and Wang, Y.: A modeling analysis of a heavy air pollution episode occurred in Beijing, Atmos. Chem. Phys., 7, 3103–3114, 2007.

Aubinet, M.: Eddy Covariance $CO_2$ Flux Measurements in Nocturnal Conditions: An Analysis of the Problem, Ecol. Appl., 18 (6), 1368-78, 2008.

Chang, X., Wang, S., Zhao, B., Cai, S., and Hao, J.: Assessment of inter-city transport of particulate matter in the Beijing–Tianjin–Hebei region, Atmos. Chem. Phys., 18, 4843-4858, 2018.

Guo, S., Hu, M., M. L. Zamora, Peng, J., Shang, D., Zheng, J., Du, Z., Wu, Z., Shao, M., Zeng, L., Mario, J., and Zhang, R.: Elucidating severe urban haze formation in China. Proc. Natl. Acad. Sci. USA., 111, 17373-17378, 2014.

Liu, X., Li, J., Qu ,Y., Han, T., Hou, L., Gu, J., Chen, C., Yang, Y., Liu, X., Yang, T., Zhang, Y., Tian, H., and Hu, M.: Formation and evolution mechanism of regional haze: A case study in the megacity Beijing, China. Atmos. Chem. Phys., 13, 4501-4514, 2013.

Mahrt, L.: Variability and Maintenance of Turbulence in the Very Stable Boundary Layer, Bound.-Layer Meteor., 135(1),1-18, 2010.

Streets, D. G., Fu, J. S., Jang, C. J., Hao, J., He, K., Tang, X., Zhang, Y., Wang, Z., Li, Z., Zhang, Q., Wang, L., Wang, B., and Yu, C.: Air quality during the 2008 Beijing Olympic Games, Atmos. Environ., 41, 480–492, 2007.

Vickers, D. and Mahrt, L.: A Solution for Flux Contamination by Mesoscale Motions With Very Weak Turbulence, Bound. Layer Meteor., 118 (3), 431-447, 2006.

Wang, Z., Li, J., Wang, Z., Yang, W., Tang, X., Ge, B., Yan, P., Zhu, L., Chen, X., Chen, H., Wand, W., Li, J., Liu, B., Wang, X., Wand, W., Zhao, Y., Lu, N., and Su, D.: Modeling study of regional severe hazes over mid-eastern China in January 2013 and its implications on pollution prevention and control, Sci. China Earth-Sci., 57, 3–13, 2014.

Zheng, G. J., Duan, F. K., Su, H., Ma, Y. L., Cheng, Y., Zheng, B., Zhang, Q., Huang, T., Kimoto, T., Chang, D., U., Cheng, Y. F. and He, K. B.: Exploring the severe winter haze in Beijing: The impact of synoptic weather, regional transport and heterogeneous reactions, Atmos. Chem. Phys., 15(6), 2969-983, 2015.

Zhong, J., Zhang, X., Dong, Y., Wang, Y., Wang, J., Zhang, Y. and Che, H.: Feedback effects of boundary-layer meteorological factors on cumulative explosive growth of $PM_{2.5}$ during winter heavy pollution episodes in Beijing from 2013 to 2016, Atmos. Chem. Phys., 18, 247-258, 2018.

Zhong, J., Zhang, X., Wang, Y., Sun, J., Zhang, Y., Wang, J., Tan, K., Shen, X., Che, H., Zhang, L., Zhang, Z., Qi, X., Zhao, H., Ren, S., Li, Y.: Relative Contributions of Boundary-Layer Meteorological Factors to the Explosive Growth of PM2.5 during the Red-Alert Heavy Pollution Episodes in Beijing in December 2016, J. Meteor. Res., 31 (5),809-819, 2017.

---

## Author Comment (AC2) · 17 Dec 2018

Dear reviewer,

We really appreciate your revealing questions and comments. These constructive opinions help to improve our work to a great extent. We did our best to respond to these comments one by one. We hope the reviewer would approve of our following response. Details are listed as follows

The authors investigated the influence of turbulence structure and urbanization on the heavy haze pollution process, and did a very detailed analysis. The results provide valuable information on the interaction between pollution and turbulence flux and the differences between urban and rural area during heavy haze time. However, further study according to following comments is needed:
Major comments:
1) the authors mentioned "the impact of the pollution process on suburb areas is much greater than that on urban areas. Urbanization seems to help reduce the consequences of pollution" in the end of abstract. What are the dynamic/physical mechanism behind that? What is the difference in PM2.5 source at urban areas and suburb areas?

**Response:** We really appreciate your questions. We want to introduce more details about these two sites and try to explain the second question before answering the dynamic/physical mechanism behind the sentence. Suburban site is located at locally flat underlying surface, as shown in Fig.1, 20 km away from the urban site. At this distance, we believe that the impact of the large-scale weather situation on both sites is consistent. Recent studies have shown that Beijing's winter pollution process can be divided into pollutant transport stage and accumulation stage (Zheng et al., 2015a; Liu et al., 2013; Guo et al., 2014; Zhong et al. 2017; Zhong et al., 2018). Some studies have shown that emissions from outside Beijing can contribute 28–70% of the ambient $PM_{2.5}$ concentrations in Beijing (An et al., 2007; Streets et al., 2007; Z. Wang et al., 2014; Chang et al. 2018). During the period we are concerned, as shown in Fig. 2, Zhong et al.'s research indicates that the pollution process is composed of the transportation of pollutants by southerly wind and accumulation of pollutants under the static wind (wind speed less than 2 m/s). Pollutant transport is dominated by large-scale weather processes, so the contribution to the source of $PM_{2.5}$ due to transport can be considered to be substantially consistent between the two sites separated by 20 km. In the accumulation stage of pollutants, it can be seen that the wind speed is less than 2 m/s in the layer below 1 km, and there is no longer large-scale transportation. At this time, the contribution of the local source may be highlighted. Due to the regulatory measures such as factory shutdowns implemented by the government in the fall of 2016 ("Bulletin on the State of China's Ecological Environment in 2016"), there are few major industrial sources in Beijing during our research phase, and there are only a small number of residential sources. In fact, in the government regulation in 2017, the residential sources had also been strictly controlled. Although the suburban site is located in a relatively flat farmland, it is still in the vicinity of the Changping county. Therefore, the residential sources between the two sites may be different, but this difference is not significant compared to the

large number of sources in the transportation process. In summary, we think that the sources of PM2.5 of the two sites are generally consistent.

[Figure]

**Figure 1** Google earth map of the suburban site within a range of approximately 1 km.

[Figure]

**Figure 2** Temporal variations in PM$_{2.5}$, PLAM, and vertical distributions of meteorological factors from 1 December 2016 to 10 January 2017 by Zhong et al. (2017), their Fig.1. (a) PM$_{2.5}$ mass concentration, (b) wind vector and wind velocity (shading; m s$^{-1}$), (c) temperature (shading; ℃), (d) RH (shading; %), and (e) PLAM. Red boxes correspond to original/transport explosive growth processes, while green boxes correspond to subsequent/cumulative explosive

growth processes.

Due to the close distance between the two stations, the large-scale weather background is consistent, and the underlying surface of the suburban site is locally flat, so the flat terrain of the suburb is used as a reference. We consider the turbulence structure over the flat surface of the suburban area as the normal state in this area under such weather and pollution source conditions, then the urban turbulent structure is influenced by urbanization. Next, we want to discuss the dynamic mechanism behind this sentence from two aspects. First, the LIST (intermittent intensity) at the suburban site is smaller (stronger) than that at the urban site (as shown in Fig.7 and Fig.8 in the manuscript). The value of $V_{smeso}(V_{turb})$ at the urban site is slightly smaller (larger) than that at the suburban site. All of these mean that the proportion of turbulence in the acquired signal at the urban site is stronger than that at the suburban site which was caused by rough underlying surface mainly. So from the turbulence intermittency perspective, the value of the LIST at the urban site is significantly greater than that at the suburban site, which means that the dynamic effect of the underlying surface of the urban area is stronger than that of the flat underlying surface of the suburban area. That is correspond to the previous studies on atmospheric turbulence over cities, such as, Roth (2000) concluded that there was an intense shear layer near the top of the city canopy which provided increased mechanical mixing in his review paper. Second, the change in turbulent flux during the pollution process also showed a difference in turbulence between the two sites. As shown in Fig.11 in the manuscript, the sensible heat flux, latent heat flux, momentum flux, and TKE in the urban and suburban sites are all weaken when pollution occurs. More importantly, the reduction of turbulence exchange during the pollution period in suburban sites is greater, compared to urban sites. In other words, the impact of the pollution process on the suburbs is much greater than that on the urban area. As mentioned earlier, we consider the turbulent structure over the suburb site as the normal state in this area under such weather and pollution conditions, the change in the underlying surface of the urban site (the dynamic effect of complex terrain and the heat island effect caused by human activities) leads to a greater resistance to the weakening effects of turbulent exchange caused by pollution. To sum up two points, the change in turbulence structure brought about by urbanization has a weakening effect on the consequences caused by pollution.
The weaker turbulence intermittently caused by urbanization provides stronger turbulent flow during heavy pollution, so from the perspective of turbulent structure, the complexity of the urban underlying surface provides better turbulent transfer conditions compared to flat underlying surface.
However, maybe this sentence is somewhat misleading when it appears alone. We changed the expression to "The turbulent effects caused by urbanization seems to help reduce the consequences of pollution under the same weather and pollution source condition, because the turbulence intermittency is weaker and the reduction of turbulence exchange is smaller over urban underlying surface." in the abstract.

According to your comments, the following changes were made:
Figure 1 in the manuscript has been modified to reflect the condition of the underlying surface around the two sites. The new Fig. 1 for the manuscript is shown as Fig. 3 here:

[Figure]

**Figure 3 Figure 1 in the manuscript**: Google Earth map of the observation sites in Beijing: (a) the observation site located in the urban underlying surface region (marked by the red pentagram) and the observation site located in the suburban underlying surface region with a flat terrain (marked by the red circle). The corresponding terrains (within a range of approximately 1 km) around the observation sites are shown in (b) and (c), respectively.

Details of the suburban site are added on page 3, lines 29-30:
"Data over a locally flat underlying surface were collected at a continuous measurement site (40.16° N, 116.28° E) in the Beijing suburb. The observational site was set up in the middle of a vast and horizontal farmland, in the vicinity of the Changping county."
We added a description of the location relationship between the two sites on page 4, lines 20-32:
"Suburban site is 20 km away from the urban site. At this distance, the large-scale weather background is consistent. As flat terrain of the suburban site, it was used as a reference. The sources of $PM_{2.5}$ of the two sites are generally consistent."
We changed the description in the abstract on the first page, line 25-28:
"The turbulent effects caused by urbanization seems to help reduce the consequences of pollution under the same weather and pollution source condition, because the turbulence intermittency is weaker and the reduction of turbulence exchange is smaller over urban underlying surface."
The last paragraph of the conclusion was reorganized, on page 26, lines 12-19:
"Due to the close distance between the two stations, the large-scale weather

background is consistent, and the suburban site is locally flat, so the turbulent structure over the flat surface of the suburban area is considered as the normal state in this area under such weather and pollution source conditions, then the urban turbulent structure is influenced by urbanization. More importantly, the reduction of turbulence exchange during the pollution period in suburban sites is greater, compared to urban sites. In other words, the impact of the pollution process on the suburbs is much greater than that on the urban area. The change in the underlying surface of the urban site (the dynamic effect of complex terrain and the heat island effect caused by human activities) leads to a greater resistance to the weakening effects of turbulent exchange caused by pollution."

2) The heights of two observational sites are different, one at 25m and the other at 10m. What are differences of temperature, wind speed, specific humidity and PM2.5 caused by the height difference?
The authors used one point to represent the whole Beijing, how much is the difference in surface temperature, wind speed, specific humidity and PM2.5 at different locations of urban Beijing?

**Response:** Thanks for your question. The problem you mentioned does exist. However, this manuscript focuses on the difference of the turbulent structure and exchange between surface and atmosphere between two sites which have different underlying surfaces and are not far apart under control of the same weather and pollution source background, the meteorological elements are only the background for the process of pollution and not the focus of this work. In the manuscript, the Fig. 2 was only used to reflect the overview of the pollution process in two sites. The sentence which describes the difference between two sites is on page 6, line 15-17. Considering the present study of turbulence characteristics, we have deleted this sentence. While for turbulence flux in the two sites, they are comparable because in our work both sites were thought to be located within the surface layer, i.e. the constant flux layer.

The structure of the atmospheric boundary layer on different underlying surfaces is shown in Fig. 4. In urban boundary layers, the constant flux layer is generally located above the urban canopy (Roth, 2000; Wood 2010;). It is generally believed that the turbulent flux in the surface layer varies little with height, and M-O similarly is applicable (Arnfield, 2003). Roth (2000) provided a comprehensive, critical review of turbulence observations over cities, which included many experiments (Kato et al. 1992; Oikawa, 1993; Oikawa and Meng, 1995; Oikawa et al., 1995; Casadio et al. 1996; Xu et al., 1996; Feigenwinter et al.,1999) that the height of the observation is above the height of dominant roughness elements. The same is true of research on urban boundary layers in recent years (Weber et al., 2010; Fortuniak et al. 2013; Nordo et al. 2013; Ao et al., 2016;). The height of the urban observation site is 25 m. As shown in Fig.5, the buildings around the site are neat and uniform, average building height is 20 m. So the urban site is thought to be located within the surface layer. The suburban site is locally flat and its observation height is 10 m, which is also located in the surface layer. Because of

the constant flux assumption in surface layer, the turbulence characteristics between the two sites is comparable. For meteorological elements, although both sites are within the surface layer, the height of observation will have an inevitable impact on their value. However, the differences in meteorological elements brought by different observational heights are not the focus of this manuscript, and have no impact on the conclusions.

It is unreasonable to use one point to represent the whole Beijing, there are differences in meteorological elements between different stations due to differences in underlying surfaces. The buildings around the urban site are evenly distributed, and the observations at the site can represent the typical underlying surface conditions of this part of the city, and the comparison with the flat underlying surface of the suburbs is also effective.

[Figure]

**Figure 4** Schematic structure of the atmospheric boundary layer (ABL) that comprises internal boundary layers between dominant surface-cover types (lake, forest, suburban, urban; not to scale) by Nordbo (2012), their Fig. 3. RBL { rural boundary layer, UBL{urban boundary layer, RSL{roughness sublayer, ISL{inertial sublayer, UCL{urban canopy layer. Logarithmic wind profiles in the near-neutral surface layers (orange) are shown with the displacement height ($z_d$) and roughness length ($z_0$) over the urban surface. The urban boundary-layer structure is adapted partly from Oke (1987).

[Figure]

**Figure 5** Google earth map of the urban site within a range of approximately 1 km.

According to your comments, the following changes were made:

A little detail about the suburban site is added on page 3, line 30:

"The EC system was mounted at a height of 8.3-m above ground which was located within the surface layer"

Details of the urban site are added on page 4, lines 9-10:

"The buildings around the site are neat and uniform, average building height is 20 m. So observations at the urban site are located within the surface layer."

A description of the turbulence quantities for both sites is added on page 4, lines 23-24:

"Since the observations at both sites are located in the surface layer, i.e. the constant flux layer, the values of turbulence flux are comparable."

The sentence which describes the difference between two sites is on page 6, line 15-17, we have deleted this sentence because the comparison between two sites is meaningless and has no impact on our work. The sentence is shown as follows:

"The horizontal wind speed, virtual temperature and water vapor at the suburban site are greater, lower, and higher than those at the city site, respectively."

3) The authors pointed out the differences between clear and pollution days in the mass and energy exchanges at surface and concluded that the pollution inhibits the mass and energy exchange at surface. However, weak vertical mixing and surface wind can cause pollutants to accumulate and lead to heavy pollution.

**Response:** You're right that weak vertical mixing and surface wind can cause pollutants to accumulate and lead to heavy pollution. In fact, the relationship between the turbulence exchange and pollution accumulation is a kind of positive feedback. In the early stage of pollution, with the cooperation of the weather system, the southerly wind transports pollutants from the industrial area to the Beijing area

(Zheng et al., 2015a; Liu et al., 2013; Guo et al., 2014). In the process of gradual development of pollution, the weather system is stagnant, and the southerly wind transport is weakened or even disappeared. At this time, the small wind condition causes the weakening of the turbulent dynamic effect, and due to the accumulation of pollutants in the initial stage of pollution, the high aerosol loading greatly cool the surface by reducing surface insolation caused by scattering and absorption of sunlight in the atmosphere (Forkel et al., 2012; Ding et al., 2016; Petaja et al., 2016; Zhong et al., 2018). Therefore, the turbulent thermal effect is weakened, the inversion layer develops, and turbulent exchange is suppressed. The weakening of turbulent exchange has caused further accumulation of pollution. The accumulation of pollutants and the weakening of turbulent exchange are positive feedback processes that interact and promote each other before the static weather situation is broken.

Minor comments:
1)Figure 2a only has one line, but 2b and 2c have two lines. Figure 2a should have two line to show the results from two stations.

**Response:** Thank you for your comments. The suburban site did not have observations of particulate matter. The observations of $PM_{2.5}$ at urban sites are used to represent the evolution of the entire pollution process, as this manuscript focuses on pollution processes rather than specific values. As mentioned in the answer of the first major comment, the sources of $PM_{2.5}$ of the two sites are not much different. In fact, the trends in contaminants across all environmental monitoring sites throughout the Beijing area are consistent, although there are some numerical differences. We choose three environmental monitoring stations, Changping (116.23°N, 40.22°E), Haidian (116.29°N, 39.99°E) and Daxing (116.40°N, 39.72°E), to prove that. The time series of mass concentration of $PM_{2.5}$ at the three environmental monitoring stations are shown in Fig. 6, their locations are shown in Fig. 7.

The data of $PM_{2.5}$ used in this manuscript is mainly to show the corresponding relationship between the trend of pollution development and intermittent turbulence, as shown in Fig.7 and Fig.8 in the manuscript. For the purposes of this work, the difference in the magnitude of the $PM_{2.5}$ values between the two sites does not affect the results. And because the pollution data from environmental monitoring sites have a large number of missing measurements during the study period, so we still use the observations from the urban site to represent the evolution of the entire pollution process.

An explanation of the data problem of $PM_{2.5}$ is added to Section 2, page 4, lines 21-22:

"The observations of $PM_{2.5}$ at urban sites are used to represent the evolution of the entire pollution process, as this study focuses on pollution processes rather than specific values."

[Figure]

**Figure 6** The time series of mass concentration of PM$_{2.5}$ at the three environmental monitoring stations in Beijing.

[Figure]

**Figure 7** Google Earth map of the observation sites in Beijing. The locations of the three environmental monitoring stations are marked by the red pentagram. The urban and suburban sites in manuscript are marked by red circle.

**References**

An, X., Zhu, T., Wang, Z., Li, C., and Wang, Y.: A modeling analysis of a heavy air pollution episode occurred in Beijing, Atmos. Chem. Phys., 7, 3103–3114, 2007.

Ao, X.Y., Grimmond, C. S. B., Chang, Y., Liu, D., Tang, Y., Hu, P., Wang, Y., J. Zou and Tan J.: Heat, water and carbon exchanges in the tall megacity of Shanghai: challenges and results. Int. J. Climatol., 36(14), 4608 – 4624, 2016.

Arnfield, A.J.: Two decades of urban climate research: A review of turbulence, exchanges of energy and water, and the urban heat island. International Journal of Climatology, 23(1):1–26, 2003.

Casadio, S., Di Sarra, A., Fiocco, G., Fuii, D., Lena, F. and Rao, M. P. : Convective characteristics of the nocturnal urban boundary layer as observed with Doppler sodar and Raman lidar. Boundary-Layer Meteorol., 79, 375-391, 1996.

Chang, X., Wang, S., Zhao, B., Cai, S., and Hao, J.: Assessment of inter-city transport of particulate matter in the Beijing–Tianjin–Hebei region, Atmos. Chem. Phys., 18, 4843-4858, 2018.

Ding, A. J., Huang, X., Nie, W., Sun, J. N., Kerminen, V.M., Petäjä, T., Su, H., Cheng, Y. F., Yang, X.Q., Wang, M. H., Chi, X. G., Wang, J. P., Virkkula, A., Guo, W. D., Yuan, J., Wang, S. Y., Zhang, R. J., Wu, Y. F., Song, Y., Zhu, T., Zilitinkevich, S., Kulmala, M. & Fu, C. B.: Enhanced haze pollution by black carbon in megacities in China. Geophysical Research Letters, 43(6), 2873-2879, 2016.

Feigenwinter, Ch., Vogt, R. and Parlow, E. Vertical structure of selected turbulence characteristics above an urban canyon. Theo,: and Appl. Climatol., 62,51-63. 1999.

Fortuniak, K.,Włodzimierz Pawlak,Mariusz Siedlecki. Integral Turbulence Statistics Over a Central European City Centre, Boundary-Layer Meteorol, 146, 257–276, 2013.

Forkel, R., Werhahn, J., Hansen, A.B., McKeen, S., Peckham, S., Grell, G., Suppan, P.: Effect of aerosol-radiation feedback on regional air quality – A case study with WRF/Chem. Atmos. Environ. 53, 202-211, 2012.

Guo, S., Hu, M., M. L. Zamora, Peng, J., Shang, D., Zheng, J., Du, Z., Wu, Z., Shao, M., Zeng, L., Mario, J., and Zhang, R.: Elucidating severe urban haze formation in China. Proc. Natl. Acad. Sci. USA., 111, 17373-17378, 2014.

Kato, N., Ohkuma, T., Kim, J. R., Marukawa, H. and Niihori, Y. Full scale measurements of wind velocity in two urban areas using an ultrasonic anemometer. J. Wind Eng. Ind. Aerodyn., 41-44,67-78, 1992

Liu, X., Li, J., Qu ,Y., Han, T., Hou, L., Gu, J., Chen, C., Yang, Y., Liu, X., Yang, T., Zhang, Y., Tian, H., and Hu, M.: Formation and evolution mechanism of regional haze: A case study in the megacity Beijing, China. Atmos. Chem. Phys., 13, 4501-4514, 2013.

Nordbo, A.: Extending the applicability of the eddy-covariance flux-measurement technique. Helsingin yliopiston verkkojulkaisut.2012.

Nordbo, A., Järvi, L., Haapanala, S., Moilanen, J. and Vesala, T.: Intra-City Variation in Urban Morphology and Turbulence Structure in Helsinki, Finland, Bound.-Layer Meteor., 146 (3),469-496, 2013.

Oikawa, S. Vertical turbulence structure in an above the urban canopy. J. Jpn. SOC. Air Pollution, 28,348-358 (in Japanese). 1993.

Oikawa, S. and Meng, Y. Turbulence characteristics and organized motion in a suburban roughness sublayer. Boundary-Layer Meteorol., 74,289-3 12. 1995.

Oikawa, S., Meng, Y., Uehara, K. and Ohara, T. A field study on diffusion around a model cube in a suburban canopy. J. Jpn SOC. Air Pollution, 30,5948 (in Japanese). 1995.

Oke, T. R.: Boundary Layer Climates.  Routledge, London, 2nd edition.1987.

Petäjä, T., Järvi, L., Kerminen, V.M., Ding, A.J., Sun, J.N., Nie, W., Kujansuu, J., Virkkula, A., Yang, X.Q., Fu, C.B., Zilitinkevich, S., Kulmala, M., 2016. Enhanced air pollution via aerosol-boundary layer. feedback in China. Sci. Rep. 6, 18998.

Roth, M.: Review of atmospheric turbulence over cities. Quarterly Journal of the Royal Meteorological Society, 126(564), 941–990, 2000.

Streets, D. G., Fu, J. S., Jang, C. J., Hao, J., He, K., Tang, X., Zhang, Y., Wang, Z., Li, Z., Zhang, Q., Wang, L., Wang, B., and Yu, C.: Air quality during the 2008 Beijing Olympic Games, Atmos. Environ., 41, 480–492, 2007.

Wang, Z., Li, J., Wang, Z., Yang, W., Tang, X., Ge, B., Yan, P., Zhu, L., Chen, X., Chen, H., Wand, W., Li, J., Liu, B., Wang, X., Wand, W., Zhao, Y., Lu, N., and Su, D.: Modeling study of regional severe hazes over mid-eastern China in January 2013 and its implications on pollution prevention and control, Sci. China Earth-Sci., 57, 3–13, 2014.

Weber S and Klaus K.: Comparison of atmospheric turbulence characteristics and turbulent fluxes from two urban sites in Essen, Germany, Theor Appl Climatol,102, 61–74, 2010.

Wood, C. R., Lacser, A., Barlow, J. F., Padhra, A., Belcher, S.E., Nemitz, E., Helfter, C., Famulari, D. and Grimmond, C.S.B.: Turbulent flow at 190 m height above London during 2006–2008: a climatology and the applicability of similarity theory. Boundary-layer meteorology, 137(1), 77-96, 2010.

Xu, Y., Zhou, C., Li, Z. and Zhang, W.: Turbulent structure and local similarity in the tower layer over the Nanjing area. Boundary- Layer Meteoml., 82, 1-21, 1997.

Zheng, G. J., Duan, F. K., Su, H., Ma, Y. L., Cheng, Y., Zheng, B., Zhang, Q., Huang, T., Kimoto, T., Chang, D., U., Cheng, Y. F. and He, K. B.: Exploring the severe winter haze in Beijing: The impact of synoptic weather, regional transport and heterogeneous reactions, Atmos. Chem. Phys., 15(6), 2969-983, 2015.

Zhong, J., Zhang, X., Dong, Y., Wang, Y., Wang, J., Zhang, Y. and Che, H.: Feedback effects of boundary-layer meteorological factors on cumulative explosive growth of $PM_{2.5}$ during winter heavy pollution episodes in Beijing from 2013 to 2016, Atmos. Chem. Phys., 18, 247-258, 2018.

Zhong, J., Zhang, X., Wang, Y., Sun, J., Zhang, Y., Wang, J., Tan, K., Shen, X., Che, H., Zhang, L., Zhang, Z., Qi, X., Zhao, H., Ren, S., Li, Y.: Relative Contributions of Boundary-Layer Meteorological Factors to the Explosive Growth of PM2.5 during the Red-Alert Heavy Pollution Episodes in Beijing in December 2016, J. Meteor. Res., 31 (5),809-819, 2017.

---

## Referee Report (RR1)

The authors' replies solve most of my concerns; however, it will be great that if the authors can have more discussions on the spatial distributions and time variations of pollutants over Beijing area (see Figures 1 and 2 obtained from aqicn.org website on Dec. 28, 2018). One station observation sometime may not represent the whole area. Anyway, in my point of view, the paper can be accepted for publication.

[Figure]

Figure 1, Spatial distribution of AQI over Beijing area

[Figure]

Figure 2, time variation of different pollutants.